# Intercomparison of Aerosol Optical Depths from four reanalyses and their multi-reanalysis-consensus

Peng Xian[1], Jeffrey S. Reid[1], Melanie Ades[2], Angela Benedetti[2], Peter R. Colarco[3], Arlindo da Silva[3], Tom F. Eck[3, 4], Johannes Flemming[2], Edward J. Hyer[1], Zak Kipling[2], Samuel Rémy[5], Tsuyoshi Thomas Sekiyama[6], Taichu Tanaka[7], Keiya Yumimoto[8] and Jianglong Zhang[9]

[1]Naval Research Laboratory, Monterey, CA, USA.
[2]European Centre for Medium-Range Weather Forecasts, Reading, UK.
[3]NASA Goddard Space Flight Center, Greenbelt, MD, USA.
[4]University of Maryland Baltimore County, Baltimore, MD, USA
[5]HYGEOS, Lille, France
[6]Meteorological Research Institute, Japan Meteorological Agency, Tsukuba, Japan
[7]Information Infrastructure Department, Japan Meteorological Agency, Tokyo, Japan
[8]Research Institute for Applied Mechanics, Kyushu University, Kasuga, Japan
[9]Department of Atmospheric Sciences, University of North Dakota, Grand Forks, ND

Corresponding author: Peng Xian (peng.xian@nrlmry.navy.mil)

Key Points:

1. Four global aerosol reanalyses are intercompared and verified with observations for their skill in simulating aerosol optical depth.
2. The study identifies the strength of each reanalysis and the regions where there are notable differences and challenges.
3. The multi-reanalysis-consensus, based on the four reanalyses, consistently ranks as one of the best regionally and globally.

**Abstract**

The emergence of aerosol reanalyses in recent years has facilitated a comprehensive and systematic evaluation of Aerosol Optical Depth (AOD) trends and attribution over multi-decadal timescales. Notable multiyear aerosol reanalyses currently available include NAAPS-RA from the U.S. Naval Research Laboratory; the NASA MERRA-2; JRAero from the Japan Meteorological Agency (JMA); and CAMSRA from Copernicus/ECMWF. These aerosol reanalyses are based on differing underlying meteorology models, representations of aerosol processes, and data assimilation methods and treatment of AOD observations. This study presents the basic verification characteristics of these four reanalyses versus both AERONET and MODIS retrievals in monthly AOD properties and identifies the strength of each reanalysis and the regions where divergence and challenges are prominent. Regions with high pollution and often mixed fine-coarse mode aerosol environments such as South Asia, East Asia, Southeast Asia, and the Maritime

Continent pose significant challenges, as indicated by higher monthly AOD root mean square
error. Moreover, regions that are distant from major aerosol source areas, including the polar
regions and remote oceans, exhibit large relative differences in speciated AODs and fine-mode vs
coarse-mode AODs among the four reanalyses. To ensure consistency across the globe, a multi-
reanalysis-consensus (MRC, i.e. ensemble mean) approach was developed similar to the
International Cooperative for Aerosol Prediction Multi-Model Ensemble (ICAP-MME). Like the
ICAP-MME, while the MRC does not consistently rank first among the reanalyses for individual
regions, it performs well by ranking first or second globally in AOD correlation and RMSE,
making it a suitable candidate for climate studies that require robust and consistent assessments.

Keywords: Aerosol, Reanalysis, Aerosol Optical Depth, intercomparison, ICAP-MME

Short Summary
The study compares and evaluates monthly aerosol optical depth of four reanalyses (RA) and their
consensus (i.e. ensemble mean). The basic verification characteristics of these RA versus both
AERONET and MODIS retrievals are presented. The study discusses the strength of each RA and
identifies regions where divergence and challenges are prominent. The RA consensus usually
performs very well on a global scale in terms of how well it matches the observational data, making
it a good choice for various applications.

## 1. Introduction

In recent years, global aerosol reanalyses have been developed by major operational and research centers, owing to the availability of long-record satellite remote sensing aerosol products and advancements in aerosol data assimilation and modeling. These reanalyses are based on their operational counterparts that are included in the "Core Four" members of the International Cooperative for Aerosol Prediction Multi Model Ensemble (ICAP-MME C4C; Sessions et al., 2015; Xian et al., 2019; Reid et al., 2022). The reanalyses include the Copernicus Atmosphere Monitoring Service ReAnalysis (CAMSRA; Inness et al., 2019) produced by the European Centre for Medium-Range Weather Forecasts (ECMWF); the Japanese Reanalysis for Aerosol (JRAero) (Yumimoto et al., 2017) developed by the Japan Meteorological Agency (JMA); the NASA Modern-Era Retrospective Analysis for Research and Applications, version 2 (MERRA-2; Randles et al., 2017); and the Navy Aerosol Analysis and Prediction System reanalysis (NAAPS-RA; Lynch et al., 2016) developed by the U.S. Naval Research Laboratory (NRL).

The aerosol reanalyses are similar to their operational counterparts and characterized by a high degree of independence in their underlying meteorology, aerosol sources, sinks, microphysics, and chemistry, as well as in their assimilation methods for aerosol optical depth (AOD) observations. A summary of the configurations of these four reanalyses is presented in Table 1 for general features and Table 2 for microphysical and optical treatments of different aerosol species. Notably, the use of operational Terra and Aqua Moderate Resolution Imaging Spectrometer data (MODIS Dark Target and Deep Blue; Levy et al., 2013; Hsu et al., 2013) is consistent across these reanalyses, although preprocessing treatments vary. These treatments include quality control, bias correction, and aggregation and sampling. Additionally, several other products, such as MultiAngle Imaging Spectroradiometer (MISR; Kahn et al., 2010), Advanced Very High Resolution Radiometer (AVHRR; e.g., Ignatov et al., 2002), and Advanced Along-Track Scanning Radiometer (AATSR, Popp et al., 2016) are assimilated into some of these reanalyses, although these additional remote sensing data probably have only a small impact during the MODIS era, as their data volume is small compared to MODIS. Therefore, between their underlying meteorology, physics, and data assimilation these reanalyses are characterized by a high degree of independence overall.

Like atmospheric reanalysis products, aerosol reanalysis products, whether used individually or in combination, have been employed for diverse applications. They provide comprehensive aerosol climatology and statistics to aid in understanding aerosol conditions across various regions and the world (e.g., Reid et al., 2012; Xian et al., 2020; Nignombam et al., 2021; Ohno et al., 2022; Rubin et al., 2023). They are widely used to address a multitude of scientific inquiries in the fields of aerosol radiative forcing (e.g., Randles et al., 2017; Markowicz et al., 2017; 2021a,b; Ohno et al., 2022; Zhang et al., 2023), aerosol-cloud interaction (e.g., McCoy et al., 2017; Ross et al., 2018; Eck et al., 2018), aerosol-cryosphere interaction (e.g., Khan et al., 2018, 2019; 2020; Roychoudhury et al., 2022), air quality and its impact on health (e.g., Tong et al., 2023; Cui et al., 2022; Jenwitheesuk et al., 2022; Lacima et al., 2022), biogeochemical cycles (e.g., Rahav et al., 2020; Borchardt et al., 2019; Mescioglu et al., 2019), among others. These reanalyses have been rigorously evaluated by the developing centers and various studies from different perspectives, including AOD and other aerosol optical properties, mass concentrations, and vertical distribution profiles. However, to date, no intercomparison among the four reanalyses has been conducted.

This study presents an intercomparison of the four available global aerosol reanalyses to evaluate their skill in simulating monthly average AOD. Additionally, this study includes the development of a Multi Reanalysis Consensus (MRC) product using a multi-model-consensus approach, similar to the ICAP Multi Model Ensemble (ICAP-MME; Sessions et al., 2015; Xian et al., 2019). The MRC is an ensemble mean (i.e., mathematical average) of the four individual reanalyses, with a spatial resolution of 1°x1° latitude/longitude and monthly temporal resolution. The study provides speciated AODs, fine-mode (FM), coarse-mode (CM) and total AODs at 550 nm for the period of 2003-2019 from three reanalyses, and all four reanalyses are available for the time period of 2011-2019. In addition, a companion study focuses on global and regional AOD trends derived from these reanalyses. The validation of AODs from the MRC, and the four component members, is performed using ground-based AEROsol Robotic NETwork (AERONET; Holben et al., 1998) observations, with MODIS AOD for spatial distribution evaluation. The validation results, as well as the AOD climatology and divergence of the reanalyses, are presented in Section 3. The study concludes with a summary of the findings in Section 4.

## 2. Data and Methods

This study intercompares the monthly average modal (total, FM, and CM) and speciated AOD products from four aerosol reanalyses (RA) and their consensus, and evaluates the RA AODs with AERONET and the combined MODIS Dark Target/Deep Blue retrievals (Levy et al., 2013; Hsu et al., 2013).

### 2.1 Individual product lines

Descriptions of the four reanalysis datasets, including CAMSRA, JRAero, MERRA-2, and NAAPS-RA v1, are provided in this section. Table 1 provides a summary of the basic features of the four reanalyses and the MRC used in this study. Table 2 offers a summary of the parameters employed to depict the microphysical and optical properties of aerosol species from these reanalyses. Furthermore, Table 3 samples hygroscopic enhancement factor values that influence optical property calculations due to the hygroscopic growth of particles at various relative humidity levels. In addition to utilizing different meteorological data, aerosol source data, AOD observations, and constructing aerosol species, notable differences exist even among similar species regarding treatments related to aerosol microphysics, optical properties, and water uptake ability for hydrophilic species.

### 2.1.1 CAMSRA

The Copernicus Atmosphere Monitoring Service (CAMS) Reanalysis (CAMSRA, Inness et al., 2019) is run at the European Centre for Medium-Range Weather Forecasts (ECMWF) and is a global reanalysis of atmospheric composition species, including aerosols. It builds on the previous reanalyses of the MACC project (Inness et al., 2013) and the CAMS interim reanalysis (Flemming et al., 2017). The CAMSRA is publicly available for the years 2003 to 2022 and is being continuously updated for future years.

The CAMSRA is based on the Integrated Forecasting System (IFS) used by ECMWF for
numerical weather prediction and meteorological reanalysis. Two additional modules are
incorporated into the IFS for the CAMSRA, one to calculate the processes and reactions of the
chemical species and one to represent the prognostic aerosol species. The aerosol scheme
includes prescribed and online emissions, dry and wet deposition, production of sulfate from a
gas-phase sulfur dioxide precursor, and the aging of hydrophobic organic matter (OM) and black
carbon (BC) to hydrophilic. The prescribed anthropogenic emissions come from the MACCity
inventory (Granier et al., 2011) and the biomass burning (BB) emissions from the Global Fire
Assimilation System, version 1.2 (GFASv1.2) (Kaiser et al., 2012). GFASv1.2 is a separate
system to the IFS that uses satellite retrievals of fire radiative power to produce the BB emissions
that are then input as fixed emissions to the aerosol scheme. The transport of the aerosol species
by advection, convection and diffusion is calculated using the meteorological component of the
IFS and the wind fields from the meteorology are also used as parameters to estimate the online
sea salt (Monahan et al., 1986) and dust (Ginoux et al., 2001) surface emissions. One key
difference between the CAMSRA set up of the IFS and that used for numerical weather
prediction, is that for the CAMSRA the radiative impact of aerosol particles and ozone on
meteorology is also accounted for.
The observations used in the CAMSRA for aerosols are of total AOD at 550nm. These come
from MODIS collection 6 satellite retrievals for the entire period covered by CAMSRA and from
the Advanced Along-Track Scanning Radiometer for the period 2003-2012. These AOD
observations are simultaneously assimilated with trace gas and meteorological observations
using the 4D variational data assimilation system of the IFS with a 12-hour assimilation window.
The products available from the CAMSRA include speciated AODs at a 3-hour temporal and
approximately 0.7 degrees spatial resolution, whereas monthly mean AODs at 550nm were used
in this study.
2.1.2 JRAero
The Japanese Reanalysis for Aerosol (JRAero) was developed by the Meteorological Research
Institute (MRI) of the Japan Meteorological Agency and Kyushu University using the global
aerosol transport model MASINGAR Mk-2 (Yukimoto et al., 2012) and a two-dimensional
variational (2D-Var) data assimilation method. The model uses the MRI-AGCM3 atmospheric
general circulation model, and considers major tropospheric aerosol components, including black
carbon (BC), organic carbon (OC), mineral dust, sea salt, and sulfate aerosols, and their precursors.
JRAero assimilates global AOD from a bias-corrected MODIS Level 3 AOD product provided by
the US Naval Research Laboratory (NRL) and the University of North Dakota
(http://doi.org/10.5067/MODIS/MCDAODHD.NRT.061) every 6 hours. Anthropogenic and
biomass burning emissions were estimated using the MACCity (MACC/CityZEN EU projects)
emission inventory (http://accent.aero.jussieu.fr/MACC_metadata.php) and the Global Fire
Assimilation System (GFAS) dataset (http://www.gmes-
atmosphere.eu/about/project_structure/input_data/d_fire). The reanalysis has a resolution of
TL159 (about 1.1° × 1.1°) with 48 vertical layers from the ground to 0.4 hPa. Validation results
and additional information can be found in Yumimoto et al. (2017).
2.1.3 MERRA-2
The NASA Modern-Era Retrospective Analysis for Research and Applications, version 2
(MERRA-2, Gelaro et al. 2017) is an atmospheric and aerosol reanalysis produced with the
NASA Goddard Earth Observing System (GEOS) Earth system model. Aerosol data assimilation
brings in data from the MODIS and MISR satellite sensors (after 2000) and includes AERONET
ground-based sun photometer observations (through 2014). The Goddard Chemistry, Aerosol,
Radiation, and Transport model (GOCART; Chin et al. 2000; Colarco et al. 2010) is run online
and radiatively coupled in the MERRA-2 system, and provides simulations of dust, sea salt,
sulfate, and black and organic carbon aerosol species.
Black and organic carbon are each partitioned into hydrophobic and hydrophilic modes, and a
single bulk sulfate aerosol species is carried. Dust and sea salt are partitioned into five non-
interacting size bins, with dust emissions based on the model 10-m wind speed and a topographic
source function following Ginoux et al. (2001), and sea salt emissions driven by the surface wind
friction speed modified from Gong (2003) and with a sea-surface temperature adjustment based
on Jaeglé et al. (2011). Explosive volcanic sulfur emissions are included through 2010 based on
Diehl et al. (2012), with a repeating annual cycle of degassing volcanic emissions subsequent.
Other emissions are as summarized in Table 1.
The analysis of AOD is performed on quality-controlled MODIS, MISR, and AERONET data as
described in Randles et al. (2017) and Buchard et al. (2015). The AOD analysis is performed by
means of analysis splitting, where first a 2-D analysis of AOD is performed using error
covariances derived from innovation data. Three-dimensional analysis increments for aerosol
mass concentration are then computed using the Local Displacement Ensemble (LDE)
methodology, which accommodates misplacement of the aerosol plumes due to source or
transport issues. The ensemble perturbations are generated at the full model resolution, without
the need for multiple model runs. Online quality control is performed as in Dee et al. (2001),
with observation and background errors estimated as in Dee and da Silva (1999). Randles et al.
(2017) and Buchard et al. (2017) describe the overall methodology and validation of the
MERRA-2 AOD reanalysis. For this study, monthly mean speciated AODs and total AOD at 550
nm with 0.5 degree latitude and 0.625 degree longitude spatial resolution were used.
2.1.4 NAAPS-RA v1
The Navy Aerosol Analysis and Prediction System (NAAPS, Lynch et al., 2016) is a global offline
chemical transport model developed at the U.S. Naval Research Laboratory. NAAPS simulates the
life cycles of aerosol particles and their gaseous precursors. The particle species include
anthropogenic and biogenic fine (ABF, a mix of sulfate, organic aerosols and BC from non-BB
sources), BB smoke, aeolian dust, and sea salt aerosols. The transport, hygroscopic growth of
particles, dry and wet removal processes of these particles, and emissions of wind-blown particles
are driven by the meteorological fields from the Navy Global Environmental model (NAVGEM,

Hogan, et al., 2014). Secondary organic aerosol (SOA) processes are represented with a $1^{st}$ order approximation method, in which production of SOA from its precursors is assumed to be instant and is pre-treated outside the model. Anthropogenic emissions come from the MACC inventory from ECMWF (Granier et al., 2011). BB smoke emission is derived from the Fire Locating and Modeling of Burning Emissions (FLAMBE, Reid et al., 2009), which is constructed based on the MODIS fire hot spot data. In the reanalysis version, additional orbital corrections and regional emission factors are incorporated. Aeolian dust emissions are determined based on the surface friction velocity to the fourth power, and surface erodibility, which is adopted from Ginoux et al. (2001) with regional tuning. Dust emission occurs when specific conditions related to surface wetness and friction velocity thresholds are met. The representation of sea spray process adheres to Witek et al. (2007), with sea salt emission being governed by sea surface wind conditions.

The NAAPS-ReAnalysis (NAAPS-RA) v1 (Lynch et al., 2016) is derived from NAAPS, with assimilation of quality-assured and quality-controlled MODIS (Zhang et al., 2006; Hyer et al. 2011) and MISR AOD products (Shi et al., 2011) using 2D-var data assimilation method (Zhang et al., 2008). It provides 3-D mass concentration, extinction, and 2-D 550 nm AOD from these aerosol species with 1°x1° latitude/longitude spatial and 6-hourly temporal resolution for the years 2003-2022. The BB smoke source and dust sources are regionally tuned to best match the FM and CM AODs with AERONET AODs. Aerosol wet removals within the tropical region were regulated with satellite precipitation product (Xian et al., 2009) to mitigate model's deficiency to simulate convective precipitation. The reanalysis shows similar decadal trend of AOD found in satellite products (e.g., Zhang et al., 2017) and was verified with various field campaign data (e.g., Reid et al., 2016; Atwood et al., 2017; Edwards et al., 2022; Reid et al., 2023) in addition to ground and space-based observations.

2.2 Multi-reanalysis-consensus (MRC)

The MRC product is a result of combining four individual aerosol reanalysis products described above. This method follows the multi-model-ensemble approach used by the International Cooperative for Aerosol Prediction (ICAP) and is based on the work by Sessions et al. (2015) and Xian et al. (2019). The data from each RA with spatial resolution different from 1°x1° lat/lon degree, is first projected onto the global map with 1°x1° lat/lon degree resolution using linear interpolation. Then the MRC value is determined by calculating the average of the values from the four RAs. No weighting among the RAs is applied, or the four RAs are weighted equally in deriving MRC. Regionally-weighted ensemble product based on the verification results shown here can be developed in the future. The MRC provides speciated and total AOD at 550 nm with a 1°x1° lat/lon degree and monthly resolution for the period 2003-2019. The MRC data for the period spanning from 2003 to 2010 relies on three RAs, while for the period from 2011 to 2019, it incorporates all four RAs, considering that JRAero data is only accessible starting from 2011.

Table 1. Summary of the characteristics of the aerosol reanalyses.

| | Developer | Meterology | Resolution lat x lon | DA method | Assimilated obs. | Species | Anthro. & Biogenic Emission | BB Emissions | Available time | reference |
|---|---|---|---|---|---|---|---|---|---|---|
| **CAMSRA** | ECMWF | Inline ERA5 | 0.7 x 0.7 | 4D-Var | DAQ MODIS, AATSR | BC, OM, Sulfate Dust, Sea Salt | MACCity (trend: ACCMIP +RCP8.5), monthly VOC | GFAS | 2003-present | Inness et al., 2019 |
| **MERRA-2** | NASA | Inline MERRA-2 | 0.5 x 0.6 | 2D-Var +LDE | Neural Net MODIS, MISR, AVHRR, AERONET | BC, OC, Sulfate Dust, Sea Salt | EDGAR V4.1, AeroCom Phase II | GFED before 2009, QFED after 2009 | 1980-present | Randles et al., 2017 |
| **NAAPS-RA** | NRL | Offline NOGAPS/NAVGEM | 1 x 1 | 2D-Var | DAQ MODIS, MISR | BB smoke, Dust, Sea Salt, ABF | MACCity, BOND POET, monthly SOA | FLAMBE | 2003-present | Lynch et al., 2016 |
| **JRAero** | JMA | Inline MRI AGCM3 | 1.1 x 1.1 | 2D-Var | DAQ MODIS | BC, OC, Sulfate Dust, Sea Salt | MACCity | GFAS | 2011-present | Yumimoto et al., 2017 |
| **MRC** | - | - | 1 x 1 | - | - | BB smoke, Dust, Sea Salt, ABF | - | - | 2003-present | this work |


Table 2. Parameters representing microphysical and optical properties of aerosol species from
the four aerosol reanalyses.

| Models \ Species | Microphysics (sectional size bins in radius or bulk effective radius in µm) | | | | | Optical parameters at 550nm for the corresponding size bins (single scattering albedo, mass extinction efficiency m2/g , and shape for dry particle) | | | | |
|---|---|---|---|---|---|---|---|---|---|---|
| | Dust | Sea salt | sulfate/ABF | BB smoke /OC/OM | BC | Dust | Sea salt | sulfate/ ABF | BB smoke /OC/OM | BC |
| CAMSRA | 0.03 - 0.55, 0.55 - 0.9, 0.9 - 20 | 0.03- 0.5, 0.5 -5, 5 - 20 | 0.005 - 20 | OM: 0.005 - 20 | 0.005 - 0.5 | 0.97; 2.56 0.90; 0.92 0.85; 0.42 sphere | 1.0; 0.73 1.0; 0.14 1.0; 0.04 sphere | Sulfate 1.0; 4.33 sphere | OM: 0.89; 2.76 sphere | 0.21; 9.41 sphere |
| MERRA-2 | 0.1 - 1.0, 1.0 - 1.8, 1.8 - 3.0, 3.0 - 6.0, 6.0 - 10 | 0.03 - 0.1, 0.1 - 0.5, 0.5 - 1.5, 1.5 - 5.0, 5.0 - 10 | Bulk, 0.16 | OC: Bulk 0.09 | Bulk, 0.04 | 0.96; 2.02 0.92; 0.64 0.89; 0.33 0.83; 0.17 0.77; 0.08 spheroids | 1.0; 0.73 1.0; 3.48 1.0; 0.74 1.0; 0.30 1.0; 0.10 sphere | Sulfate 1.0; 3.15 sphere | OC: 0.96; 2.67 sphere | 0.21; 9.28 sphere |
| NAAPS-RAv1 | Bulk, 2.5 | Bulk, 1.5 | Bulk, 0.14 | Smoke: Bulk, 0.17 | N/A | 0.88; 0.59 sphere | 0.99; 1.42 sphere | ABF 0.9; 3.48 sphere | Smoke: 0.89; 4.48 sphere | N/A |
| JRAero | 0.100 – 0.159, 0.159 – 0.251, 0.251 – 0.398, 0.398 – 0.63, 0.63 – 1.00, 1.00 – 1.59, 1.59 – 2.51, 2.51 – 3.98, 3.98 – 6.30, 6.30 – 10.0 | 0.100 – 0.159, 0.159 – 0.251, 0.251 – 0.398, 0.398 – 0.63, 0.63 – 1.00, 1.00 – 1.59, 1.59 – 2.51, 2.51 – 3.98, 3.98 – 6.30, 6.30 – 10.0 | Bulk, 0.15 | OC: Bulk, 0.18 | Bulk, 0.18 | 0.96; 1.78 0.98; 3.36 0.97; 3.32 0.94; 1.45 0.90; 0.82 0.86; 0.48 0.81; 0.29 0.75; 0.18 0.68; 0.11 0.61; 0.07 sphere | 1.0; 0.17 1.0; 0.56 1.0; 1.36 1.0; 1.97 1.0; 1.53 1.0; 0.54 1.0; 0.39 1.0; 0.23 1.0; 0.14 1.0; 0.08 sphere | 1.0; 2.26 sphere | 0.96; 1.60 sphere | 0.16; 5.34 sphere |



Table 3. Hygroscopic enhancement factor (*f*) at different relative humidity (RH) levels for
various aerosol species in the four RAs. In MERRA-2, $f$ for sea salt varies with size bins, thus a
range for *f* is presented here. Notably, NAAPS-RA v1 does not explicitly contain BC species.
More specific details can be found in the references provided in Table 1.

| RH (%) | Sea salt | | | | Sulfate/ABF | | | | BB smoke/OM/OC | | | | BC | | |
|---|---|---|---|---|---|---|---|---|---|---|---|---|---|---|---|
| | CAMSRA | MERRA2 | NAAPSRA | JRAero | CAMSRA | MERRA2 | NAAPSRA | JRAero | CAMSRA | MERRA2 | NAAPSRA | JRAero | CAMSRA | MERRA2 | JRAero |
| <30 | 1.00 | 1.00 | 1.00 | 1.00 | 1.00 | 1.00 | 1.00 | 1.00 | 1.00 | 1.00 | 1.00 | 1.00 | 1.00 | 1.00 | 1.00 |
| 30 | 1.00 | 1.17-1.22 | 1.00 | 1.36 | 1.00 | 1.23 | 1.00 | 1.24 | 1.00 | 1.14 | 1.00 | 1.12 | 1.00 | 1.00 | 1.00 |
| 40 | 1.44 | 1.21-1.28 | 1.07 | 1.48 | 1.17 | 1.31 | 1.08 | 1.32 | 1.17 | 1.19 | 1.03 | 1.16 | 1.00 | 1.00 | 1.00 |
| 50 | 1.56 | 1.26-1.35 | 1.17 | 1.60 | 1.22 | 1.39 | 1.18 | 1.40 | 1.20 | 1.24 | 1.06 | 1.20 | 1.00 | 1.00 | 1.00 |
| 60 | 1.67 | 1.33-1.44 | 1.29 | 1.70 | 1.28 | 1.46 | 1.32 | 1.45 | 1.30 | 1.29 | 1.11 | 1.30 | 1.00 | 1.01 | 1.00 |
| 70 | 1.80 | 1.44-1.56 | 1.48 | 1.80 | 1.36 | 1.54 | 1.53 | 1.50 | 1.40 | 1.34 | 1.16 | 1.40 | 1.00 | 1.03 | 1.00 |
| 80 | 1.99 | 1.60-1.77 | 1.78 | 2.00 | 1.49 | 1.64 | 1.87 | 1.60 | 1.50 | 1.44 | 1.25 | 1.50 | 1.20 | 1.19 | 1.20 |
| 85 | 2.13 | 1.74-1.93 | 2.03 | 2.20 | 1.58 | 1.69 | 2.16 | 1.70 | 1.55 | 1.52 | 1.32 | 1.55 | 1.30 | 1.30 | 1.30 |
| 90 | 2.36 | 1.96-2.19 | 2.45 | 2.40 | 1.73 | 1.77 | 2.65 | 1.80 | 1.60 | 1.64 | 1.42 | 1.60 | 1.40 | 1.41 | 1.40 |
| 95 | 2.88 | 2.43-2.74 | 3.37 | 2.90 | 2.09 | 1.91 | 3.74 | 1.90 | 1.80 | 1.88 | 1.61 | 1.80 | 1.50 | 1.54 | 1.50 |

## 2.3 AERONET

AERONET is a global ground-based sun photometer network managed by NASA. Sun and sky radiance at multiple wavelengths, covering the near-ultraviolet to near-infrared, are measured (Holben et al., 1998). Version 3 Level 2 AERONET daily data (Giles et al., 2019), which are cloud-screened and quality-assured, are used in this study. The estimated uncertainty in AERONET measured AOD, due primarily to calibration uncertainty, is ~0.01-0.02 at optical airmass of one for network field instruments (with the highest errors in the UV; Eck et al., 1999).

The 550 nm FM and CM AODs and total AODs are derived with the Spectral Deconvolution Method (SDA; O'Neill et al. 2003). The AERONET SDA product has been verified using in situ measurements (see for example Kaku et al., 2014). The spectral separation of FM and CM particles is determined based on their distinctive optical properties and complete size distributions. As part of this separation, a diameter of approximately 1µm serves as an approximate threshold to differentiate FM and CM particles. This optical separation is different from the sub-micron fraction (SMF) method that uses a specified cutoff radius of the particle size distribution in the AERONET (AOD & sky radiance) inversion and allows more data to be available compared to the SMF method. The FM fraction based on SDA is generally comparable and slightly greater than SMF (O'Neill et al., 2023).

This study uses AERONET sites that have more than 5 years of observations and more than 1000 daily data between 2011 and 2019 for verification purposes. Monthly AOD was derived for months that have more than 15 days of daily data. Then only sites with more than 45 total number of months (upper three quartiles of sites regarding total number of monthly data) were selected. This resulted in a total number of 200 sites globally. The list of sites along with latitude/longitude coordinates and elevation details for the studied regions is accessible in Table S1. Additionally, the locations of all sites can be identified in Figure 8.

## 2.4 MODIS AOD

Three MODIS AOD products are used as reference datasets to show global distribution of AOD climatology and the divergence among the retrieval products in comparison with the RAs. The level 3 MODIS AOD data for Dark Target (DT) were constructed using collection 6.1 Aqua MODIS level 2 DT data. The level 2 DT MODIS aerosol retrievals are available at a $10 \times 10$ km² spatial resolution over both land and ocean. These aerosol retrievals were initially averaged on a

daily basis at a spatial resolution of 0.5×0.5° lat/lon. Only data with a quality flag of "marginal"
or better were used in the analysis. Additionally, retrievals with a cloud fraction larger than 80%
were excluded to minimize cloud contamination, as suggested by Zhang et al. (2005). The level 3
DT MODIS AOD data (0.5×0.5° lat/lon) were then constructed using the daily averaged AOD
data.
Similar approaches were applied to C6.1 Aqua MODIS level 2 Deep Blue (DB) AOD data.
Unlike the MODIS DT aerosol retrievals, which are available over regions with low surface
reflectance, the DB retrievals are also available over some bright regions, such as desert regions.
No over-ocean aerosol retrievals, however, are included in the MODIS level 2 aerosol data. The
level 2 DB MODIS aerosol data were used to construct daily averages at a spatial resolution of
0.5×0.5° (lat/lon). No quality flag and cloud fraction thresholds were applied. The level 3 DB
MODIS AOD data (0.5×0.5° lat/lon) were constructed using the daily averaged AOD data.
The third MODIS AOD product is a data-assimilation-quality AOD dataset. It was based on C6.1
DT and DB retrieval products (Levy et al., 2013). Strict quality control and bias-correction
processes were applied as described in Zhang and Reid (2006) and Shi et al. (2011) for over water,
Hyer et al. (2011) for over land, and Shi et al. (2013) for over desert regions. These quality control
processes were updated for the C6.1 data and the final MODIS C6.1 AOD (550 nm) data is a level
3 product with 1°x1° lat/lon spatial and 6-hourly temporal resolution. This product has a cut-off at
40°S to filter out potential cloud-contaminated data south of this latitude. The 6-hour-averaged
AOD data were then binned into monthly means.
Note that MODIS AOD products are well known to low bias significant aerosol events (e.g., Reid
et al., 2022; Gumber et al., 2023) and slightly high bias clean environment (e.g. Wei et al., 2019),
which could affect AOD climatology to some degree.
2.5 Analysis Method
This study aims to investigate the divergence and utility of RAs for climate-scale studies by
exploring the AOD at 550 nm. To achieve this goal, the AOD data from the RAs, as well as
MODIS, were spatially and temporally binned into 1°x1° degrees and monthly resolutions. For the
purpose of verification and intercomparison analysis, only the data between 2011 and 2019 were
used as that is the period when all the RAs have data. The study focuses on the 550 nm AOD
parameter since it is available for all four aerosol RAs and MODIS. Furthermore, the AERONET
FM and CM AODs at 550 nm were obtained using the SDA method described in Sect. 2.3.
The study examines the performance of RAs globally and regionally. Sixteen regions, including
the globe, are defined for regional aerosol property analysis. They include East Asia, Southeast
Asia, South Asia, Maritime Continent, Australia, Southwest Asia, Europe, Northwest Africa,
South Africa, West North America, East North America, Central America, South America, as
indicated by the rectangular boxes in Figure 5, and Arctic (north of 70°N), and Antarctic (south of
75°S). There is no AERONET site satisfying site selection criteria as described in Section 2.3 in
the Arctic and Antarctic, so these two regions are excluded for regional verification though they
are included in other analyses.

Regarding the aerosol species, the study focuses on BB smoke, ABF in NAAPS-RA, and its equivalent of sulfate for MERRA-2, CAMSRA, and JRAero, as well as dust and sea salt. The definition of species follows the ICAP practices (Sessions et al., 2015; Xian et al., 2019) for the operational counterparts of these RAs and previous applications of these RAs (e.g., Xian et al., 2022), in which the sum of OM and BC AODs from CAMSRA, and the sum of OC and BC AODs from MERRA-2 and JRAero, is used to approximate BB smoke AODs. Although this separation of species may be somewhat arbitrary, the study takes into account the fact that different aerosol types and sources may be represented differently in each RA. For example, the NAAPS-RA model characterizes aerosol species by emission source rather than chemical speciation, which makes it unique. In contrast, CAMSRA, MERRA-2, and JRAero characterize OM or OC, BC, and inorganic species, merging contributions from various anthropogenic, biomass burning and biogenic sources.

The study also assumes that all sea salt and dust are CM, while other aerosol species are FM. The segregation of sea salt and dust to the CM category is based on the fact that only a small portion of total sea salt or dust AOD at 550nm are attributed to their FM components. For example, FM sea salt represents about 17%, 10% and 11% of total sea salt AOD globally in MERRA-2, CAMSRA and JRAero respectively. The numbers are about 30%, 39% and 32% for dust. While FM fraction of dust during dust storms in Africa varies between 20-25% according to AERONET. The FM fraction of dust from MERRA-2, CAMSRA and JRAero might be biased high as these global models tend to overestimate FM dust and underestimate CM dust (for example O'Sullivan et al., 2020; Kramer et al., 2020). In contrast, NAAPS-RA assumes all sea salt and dust are CM. Verification results based on the FM and CM AODs derived using the FM fractions of sea salt and dust from MERRA-2, CAMSRA and JRAero can be found in the supplemental material (Fig. S2-4). Generally, the validation of FM and CM AODs with AERONET data shows a degradation in performance for the three RAs compared to the verification results presented below, as discussed in section 3.3.1.

For every AERONET site, the time series of monthly modal AOD from each RA is first extracted from the model grid that encompasses the site's location. Bias, root mean square error (RMSE), and coefficient of determination ($r^2$) are then computed for each site and each RA. The regional validation outcome is derived from the average of validation statistics across all sites within the region (see Table S1 for the sites included in each region). Following the criteria for site selection outlined in section 2.3, only 200 sites are available globally, and certain regions have only a few sites (a minimum of three sites, such as in South Africa) to represent the entire region; hence, no site weighting within a region is applied. It is acknowledged that this averaging method could bias the global validation result toward regions densely populated with sites, notably North America and Europe. The AOD validation results for total, FM, and CM AOD at 550nm are presented accordingly.

3. Results

3.1 Total and speciated AOD climatology

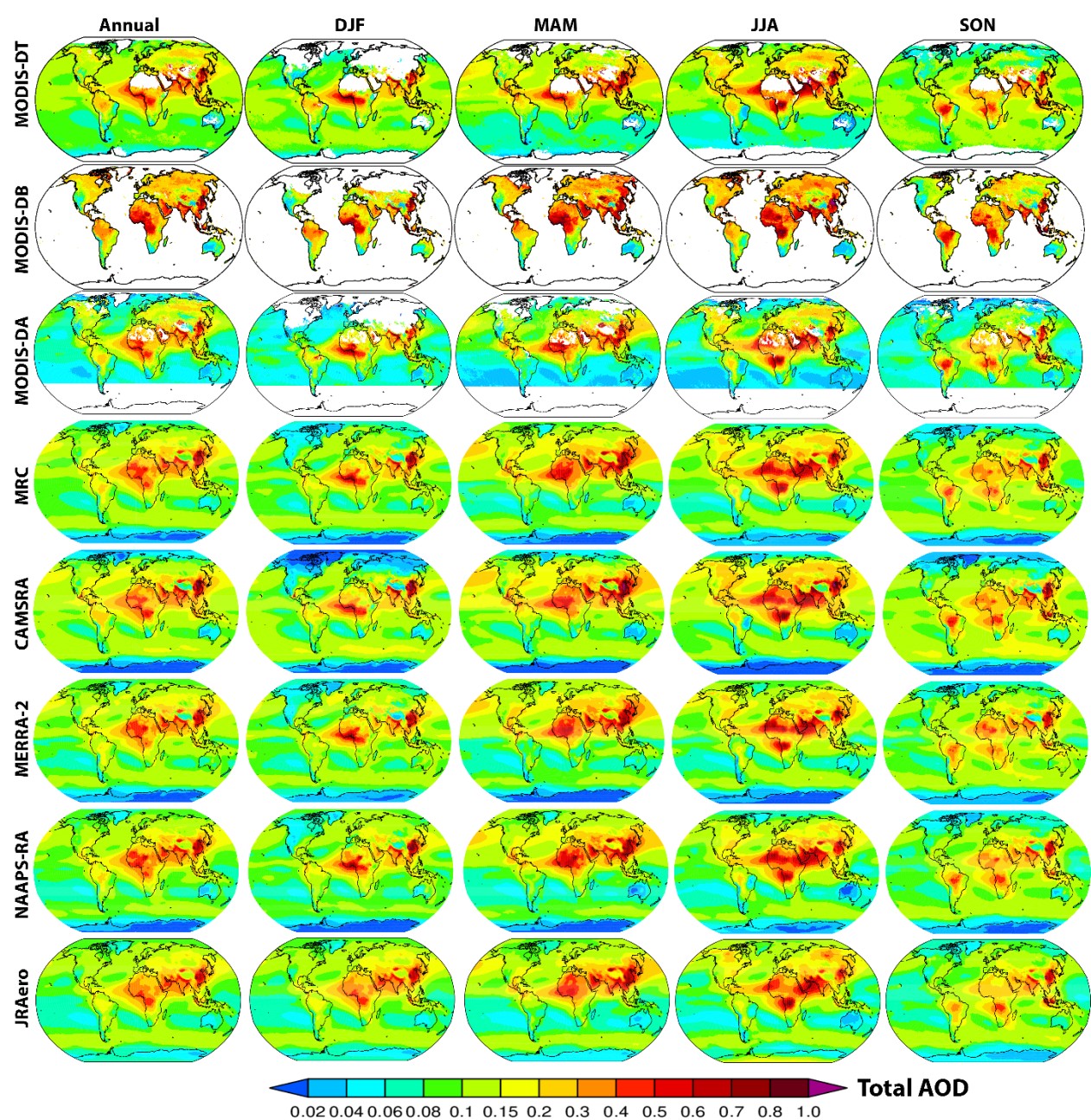

Figure 1. Annual and seasonal total 550nm AOD climatology from three MODIS products, the four RAs, and the MRC over 2003-2019, except JRAero for 2011-2019. MODIS-DA is the data-assimilation-quality AOD dataset described in Section 2.4. In the MODIS plots, the white area means a lack of data attributed to either none valid-retrievals or quality-control filtering. Notably, MODIS-DB data is only available over land.

The climatological annual and seasonal mean total AODs at 550nm from the three MODIS AOD datasets and the four aerosol RAs and the MRC are presented in Figure 1. In general, there are very similar spatial AOD distribution patterns and AOD magnitude among the RAs and MODIS

datasets for all four seasons. This is expected as MODIS total AOD is assimilated into all of these
RA products as well as used to tune the model components such as emissions. High AOD regions
include the dust-dominated Sahara in Mar-Apr-May (MAM) and Jun-Jul-Aug (JJA), Sahel in Dec-
Jan-Feb (DJF) and MAM, Southwest Asia and Taklamakan in MAM and JJA, anthropogenic
pollution-dominated East Asia and South Asia throughout the year, BB smoke-dominated South
Africa, South America in JJA and Sep-Oct-Nov (SON), Southeast Asia in MAM, Maritime
Continent in SON, and high-latitude North America and Eurasia in JJA. For the annual mean,
MODIS AODs from all the three products are relatively high compared to the MRC in the northern
hemisphere's high latitudes due to seasonal sampling bias. MODIS was able to retrieve AOD
during biomass burning active season, i.e. boreal Summer-to-Fall, but it couldn't retrieve AOD
during northern winter in the high latitudes due to the lack of sunlight and the high snow/ice
coverage. The high AOD over high-latitude Eurasia and North America in MODIS annual mean
is a general reflection of MODIS summertime AOD, which is captured by all the RAs in their
summertime mean AODs.
It is worth noting that MODIS-DB AOD generally exhibits slightly higher values compared to
MODIS-DT AOD, except in high terrain regions (e.g., Western North America). On the other
hand, MODIS-DA AOD tends to be slightly lower (approximately 0.02 magnitude) than MODIS-
DT AOD over oceanic regions due to bias-correction procedures. When compared to MODIS-DT,
AODs from the RAs tend to align more closely, especially over oceanic areas. Furthermore, RAs
typically exhibit lower AODs compared to MODIS-DB over regions affected by African and
Arabian dust. Overall, the divergence in total AOD climatology among the RAs is comparable to
or even smaller than the divergence observed in the MODIS products.

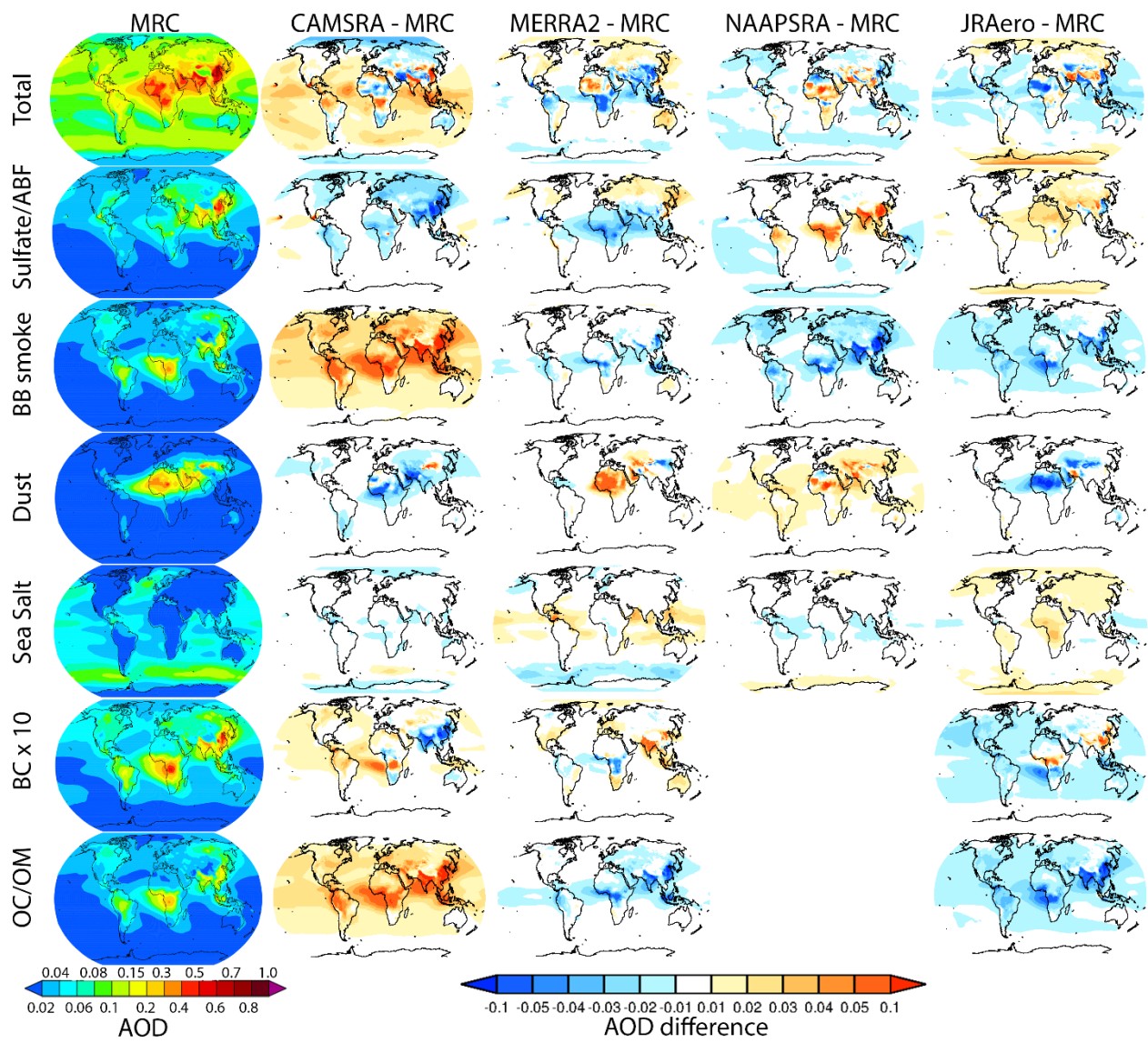

Figure 2. Annual mean total and speciated AODs of the MRC and the AOD difference between the individual RA and the MRC based on the 2011-2019 average. BB smoke is approximated as the sum of OC/OM and BC in CAMSRA, MERRA-2 and JRAero.

Previous experience with multi-model ensembles suggests that the consensus of multi-models, in general, shows better skill than individual contributing models (Sessions et al., 2015; Xian et al., 2019; Reid et al., 2022). Similar verification conclusion is also drawn in Section 3.3. Therefore, the total and speciated AODs from the MRC based on the 2011-2019 average are used as a baseline here and are shown in Figure 2. As expected, sulfate/ABF AOD is relatively high over population-dense and industrially polluted regions, dust AOD is high over major desert and arid regions, and sea salt AOD is relatively high over mid-to-high latitude oceans. BB smoke and its components BC and OC/OM are relatively high over South Africa, South America, Southeast Asia, the Maritime continent, and Siberia, North American high latitudes major BB source regions. BC and OC/OM AOD are also relatively high over South Asia and East Asia, where sources other than

BB, such as anthropogenic emission, are the main contributors, as suggested by contrasting smoke AOD contribution to the total AOD between NAAPS-RA and other RAs in these regions (Figs 3 and 10. Noting that smoke AOD is driven by BB in NAAPS-RA, while smoke AOD is a sum of BC and OC/OM from the other RAs).

Shown also in Figure 2 are the total and speciated AOD differences between the individual RA and the MRC. For total AOD, CAMSRA is apparently higher than the other three RAs over the ocean, which is consistent with the findings on its operational counterpart of high biased FM AOD verified with Maritime Aerosol Network over the ocean in Reid et al. (2022). This high bias is attributed to its universally higher OM/smoke AOD compared to other RAs, and suggests that CAMSRA may have higher BB emissions and/or higher secondary production of OM compared to the other RAs. Sulfate AOD is relatively low in CAMSRA except for some highly biased hotspots around outgassing volcanoes (in particular Mauna Loa and near Mexico City) as mentioned in Inness et al (2019). Differences in species definitions affect the comparison with NAAPS-RA: NAAPS-RA ABF AOD is higher than sulfate AOD in other RAs especially in East Asia, South Asia, central Africa, and north South America, and these deviations are counterbalanced by opposite deviations in the BB AOD. This is expected as ABF in NAAPS-RA includes additional aerosol sources besides sulfate, and some of these sources are included in the BB AOD for other models. For dust AOD, MERRA-2 is relatively higher over north Africa and the Arabian Peninsula and NAAPS-RA is relatively higher over most regions, including oceanic areas, while CAMSRA and JRAero are relatively lower over most regions except around Gobi desert for CAMSRA and Iran for JRAero. As for sea salt AOD, MERRA-2 is relatively higher over the tropical oceans, and lower over the southern ocean. JRAero sea salt AOD is relatively higher over most continents, which is probably unphysical.

The differences in speciated AOD result in significant variations in their contributions to the total AOD, as illustrated in Figure 3. For instance, the considerably higher BB smoke AOD in CAMSRA compared to other RAs makes BB smoke the predominant contributor to total AOD in the CAMSRA over most continents, adjacent water bodies, and polar regions, except for regions where dust is dominant. Sulfate AOD, on the other hand, contributes more to the total AOD, particularly over oceanic regions in the JRAero compared to other RAs. Both MERRA-2 and JRAero exhibit higher sulfate contributions along the western coasts of South America and North America, suggesting possible increased production of dimethyl sulfide (DMS) in those areas. Dust AOD, on the other hand, contributes more to the total AOD particularly over oceanic regions in NAAPS-RA compared to the other RAs. Sea salt AOD is found to contribute more to the total AOD in the high-latitude oceans and the Antarctic in NAAPS-RA compared to the other RAs. The OC/OM AOD contribution to the total AOD closely mirrors the distribution of BB smoke, as anticipated. The contribution of BC to the total AOD is generally small, ranging between 5-10% in BB regions, except for central South Africa where it reaches 10-15%. Despite the higher ratio of BB smoke AOD to total AOD in CAMSRA, the ratio of BC to total AOD over East Asia and South Asia is smaller in CAMSRA compared to MERRA-2 and JRAero, suggesting that BC emissions from anthropogenic sources maybe lower in CAMSRA (also Fig. 2). Finally, the contributions of FM and CM AOD to the total AOD are also depicted in Figure 3. It is consistent

among the RAs that FM is the dominant contributor over most land regions except for regions where dust is dominant, such as North Africa, the Arabian Peninsula, the Middle East, and the Gobi. In all the RAs, CM is the dominant contributor over oceanic regions, except for regions influenced by continental BB smoke and pollution outflow. The contribution of CM in CAMSRA is generally smaller in tropical to mid-latitude oceans compared to other RAs, due to its higher contribution from BB smoke. It is also noted that CM is dominant over FM in the Antarctic in NAAPS-RA, while FM is dominant in the Antarctic in the other three RAs, though total AOD is very small (annual and seasonal means < 0.04 from MRC) and hard to validate due to lack of observational data.

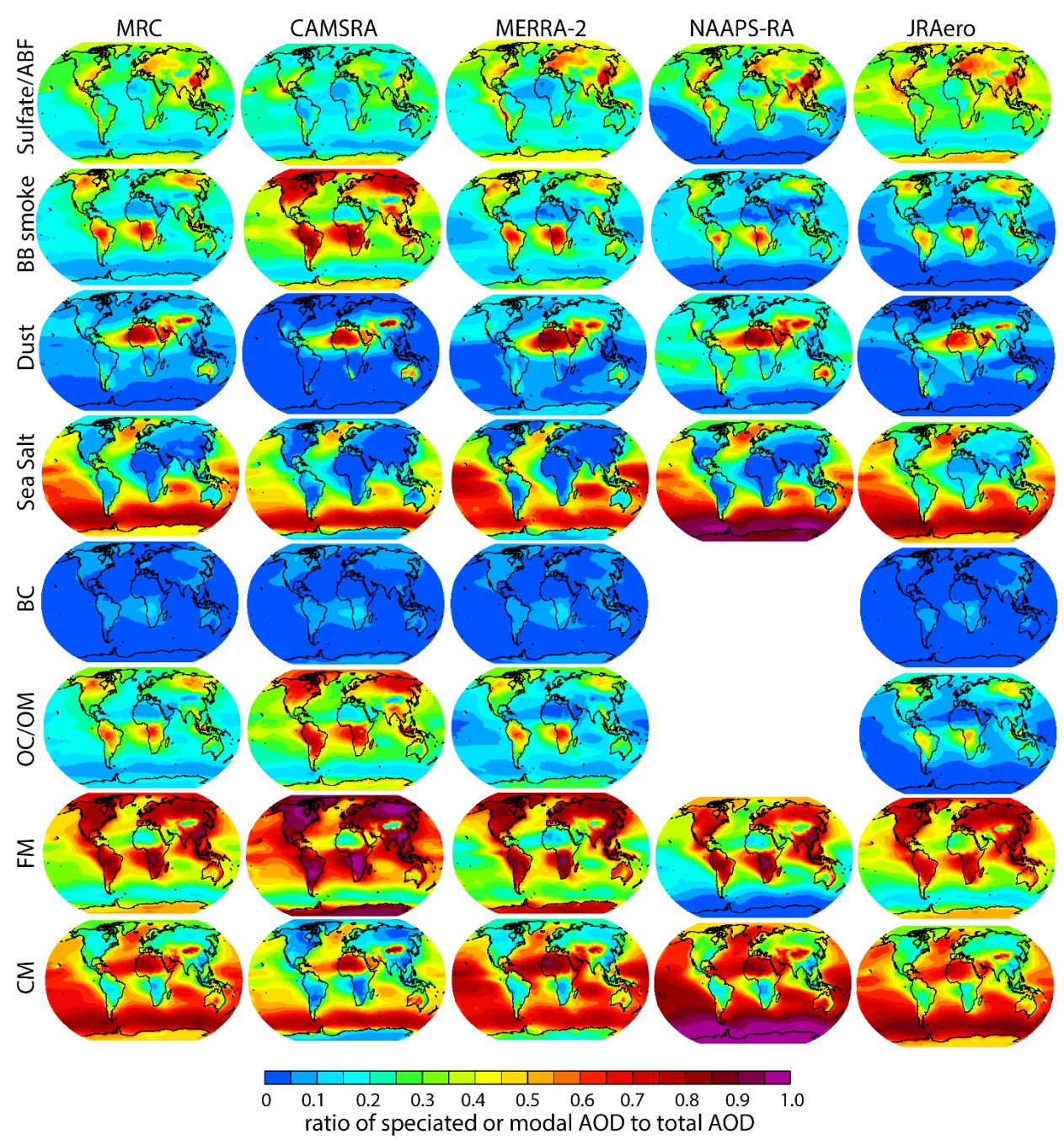

Figure 3. Ratio of speciated AODs, FM and CM AODs to total AOD from the MRC and the individual RAs based on the 2011-2019 annual average.

Table 4 provides a summary of global-average total AOD and speciated AODs, as well as the contributions of speciated AOD to total AOD for all the RAs. Overall, the annual and global mean total AODs are similar, hovering around 0.14 for most RAs. All land and ocean mean AODs are within 0.006 of the MRC with the exception of CAMSRA over ocean, which is higher than the MRC by +0.024.

Speciated AODs, especially smoke AOD and OM/OC AOD display greater divergence among the
RAs. Smoke and OM AODs from CAMSRA are 2-3 times higher than those from the other RAs.
Smoke AOD contributes to 41% of total AOD in CAMSRA, while ranging from 16%-22% in
other RAs. Moreover, the standard deviation of smoke and OM AODs with respect to the 12
months is also higher in CAMSRA than in other RAs. The contribution of dust AOD to total AOD
varies from 13% to 28% for all the RAs, with NAAPS dust AOD being the highest among the RAs
and about 2 times that of CAMSRA, which has the lowest dust AOD among the RAs. The
contribution of sulfate/ABF AOD to total AOD ranges from 23% to 34%, with the highest
contribution observed in JRAero, even larger than the ABF AOD contribution in NAAPS-RA. Sea
salt AOD contributes 25% to 35% to total AOD in the RAs with JRAero being the highest. BC
AOD, on the other hand, contributes only 3% to 4% of total AOD across the RAs. The FM's
contribution to the overall AOD varies across different datasets. In MERRA-2, NAAPS-RA, and
JRAero, FM accounts for 44% to 51% of the total AOD. However, in CAMSRA, its contribution
is notably higher at 63%, primarily due to its significant contribution from OM. Conversely, CM's
contribution to total AOD is consistent across the three RAs, ranging from 49% to 56%. In contrast,
CM's contribution is lower, at 37%, in CAMSRA.
Table 4. Global area-weighted mean modal (total, FM, CM) and speciated AOD and standard
deviation of monthly AOD based on 2011-2019 data. Percentage numbers in the brackets are
contributions of speciated AOD to total AOD. Global mean total AODs over land and water are
shown in the last two rows.

| | global mean AOD | | | | | AOD standard deviation w.r.t. 12 months | | | | |
|---|---|---|---|---|---|---|---|---|---|---|
| | CAMSRA | MERRA2 | NAAPSRA | JRAero | MRC | CAMSRA | MERRA2 | NAAPSRA | JRAero | MRC |
| total | 0.151 | 0.137 | 0.134 | 0.134 | 0.139 | 0.018 | 0.010 | 0.011 | 0.012 | 0.013 |
| dust | 0.019 (13%) | 0.029 (21%) | 0.037 (28%) | 0.021 (16%) | 0.026 (19%) | 0.008 | 0.009 | 0.009 | 0.009 | 0.008 |
| sea salt | 0.037 (25%) | 0.041 (30%) | 0.038 (28%) | 0.045 (34%) | 0.040 (29%) | 0.001 | 0.001 | 0.003 | 0.002 | 0.001 |
| sulfate/ABF | 0.034 (23%) | 0.037 (27%) | 0.037 (28%) | 0.046 (34%) | 0.039 (28%) | 0.002 | 0.001 | 0.001 | 0.002 | 0.001 |
| smoke | 0.062 (41%) | 0.030 (22%) | 0.022 (16%) | 0.022 (16%) | 0.034 (24%) | 0.009 | 0.007 | 0.007 | 0.007 | 0.007 |
| BC x 10 | 0.061 (4%) | 0.059 (4%) | - | 0.044 (3%) | 0.054 (4%) | 0.013 | 0.009 | - | 0.008 | 0.009 |
| OC/OM | 0.056 (37%) | 0.024 (18%) | - | 0.018 (13%) | 0.033 (24%) | 0.007 | 0.006 | - | 0.006 | 0.006 |
| FM | 0.096 (63%) | 0.067 (49%) | 0.059 (44%) | 0.068 (51%) | 0.073 (53%) | | | | | |
| CM | 0.056 (37%) | 0.070 (51%) | 0.075 (56%) | 0.066 (49%) | 0.066 (47%) | | | | | |
| land total | 0.180 | 0.174 | 0.175 | 0.176 | 0.176 | | | | | |
| water total | 0.136 | 0.118 | 0.112 | 0.111 | 0.112 | | | | | |


## 3.2 Geographical divergence of speciated AOD among the four RAs

The divergence of the global-average total and speciated AODs is already documented in Table 4.
Figure 4 provides the geographical distribution of the relative spread of speciated annual mean
AODs from the RAs to their means. Spread, in this context, is defined as the ratio of the standard
deviation of the RAs AODs to their mean. It is noteworthy that the relative spread of total AOD
from the four RAs is generally small, except for polar regions and specific hotspots where known
issues exist. For instance, biases in CAMSRA AOD have been identified over Hawaii and
Mexico's volcanic outgassing regions. In polar regions, there are limited satellite observations to
constrain model fields, resulting in a larger spread, which is consistent with the findings of Xian
et al. (2022) on AODs from CAMSRA, MERRA-2 and NAAPS-RA over the Arctic. Similarly,
over high terrains with snow and ice covers, such as the Himalayas and the Andes, and over desert
regions, such as the Australian deserts, and the Bodele Depression region in the Sahara, both
retrievals and models face challenges, leading to a larger spread. Moreover, over the Maritime
Continent, where high cloud coverage poses challenges to remote sensing retrievals for both AOD
and BB smoke emissions, the spread is also relatively large.
The aforementioned characteristics are also evident in the spread of speciated AODs. However,
the spreads of the speciated AODs among the RAs are much larger compared to the total AOD,
particularly in regions that are remote from aerosol sources. This suggests that the efficiency of
removal processes during long-range transport may differ. This is also relevant to the fact that data
assimilation constrains the total AOD, but speciated AOD remains unconstrained. Moreover, the
disparities in definitions of species, such as sulfate/ABF, BB smoke, OC/OM, as discussed in
Section 2.5, can also influence the spread of these FM species. The relative spread of speciated
AODs being much larger than that of total AOD, is broadly consistent with the AeroCom results,
where global climate models (without data assimilation) were intercompared in terms of aerosol
optical properties and life cycles (Kinne et al., 2006; Textor et al., 2006; Gliß et al., 2021).

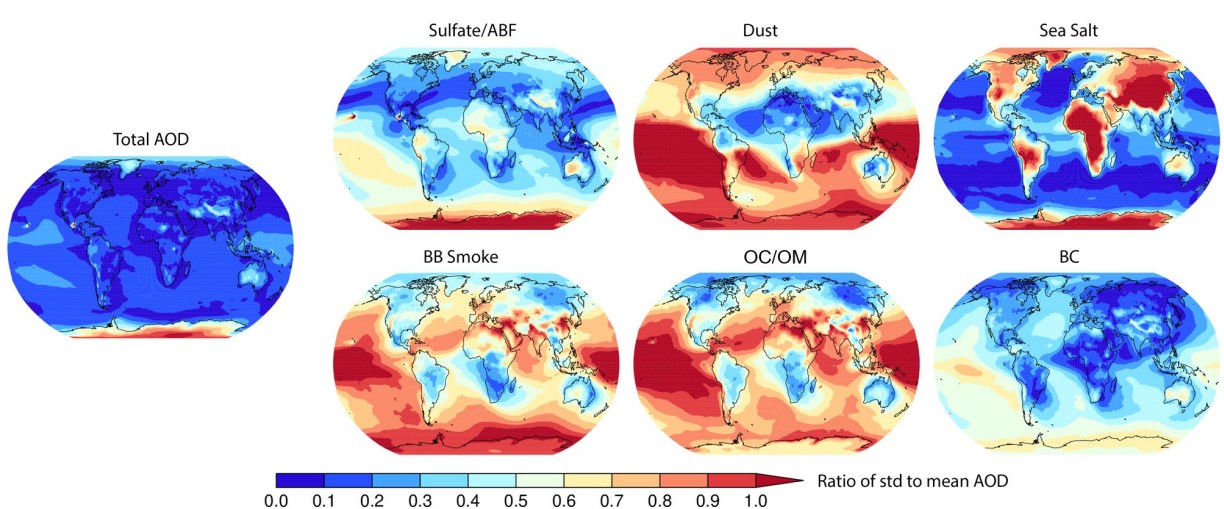


Figure 4. Spread of total and speciated climatological annual-mean AOD among the four RAs.
Spread here is defined as the ratio of the standard deviation of the RA AODs to their mean.
**3.3. Evaluation with AERONET AOD**
This section presents evaluation of the monthly performance of the four RAs plus the MRC at the
AERONET sites on regional and global scales. Both skill and consistency of the different RAs and
consensus are evaluated.
3.3.1 Bias, RMSE, and correlation between the RAs and AERONET
The regional and global mean modal AOD bias, RMSE, and coefficient of determination for the
four RAs and the MRC are shown as bar graphs on global maps in Figures 5, 6 and 7. Regarding

regional bias, all the RAs except for CAMSRA, have large negative biases (on the order of -0.1) in total AOD over Southeast Asia, South Asia, and the Maritime continent (Figure 5). The much smaller negative bias in total AOD over these regions in CAMSRA is a result of the cancelation of a positive bias in FM, possibly due to high biased OM/smoke AOD, and a negative bias in CM. The large negative biases over these regions in the other RAs are mainly attributed to large negative biases in FM AOD in general. It is also noted that CAMSRA is biased relatively high in total AOD due to high FM bias over East Asia. Over other regions and the globe, all the RAs have relatively small biases and in general slight positive biases, with CAMSRA having the largest positive bias, due mainly to relatively high OM/smoke AOD. The cancellation effect of positive FM bias and negative CM bias in CAMSRA are also visible.

Total AOD RMSEs are relatively high over all Asian regions and North Africa compared to other regions for all the RAs (Figure 6). The contribution of FM to total AOD RMSE is larger than that from CM globally, except over dust-influenced region, including North Africa and, for most models, Southwest Asia and Central America. The correlations of total AOD between the RAs and AERONET data are mostly reasonable for all the regions (Figure 7). Some relatively low-performance regions (total AOD $r^2$ less than 0.60 for at least one RA) include South Asia, Southwest Asia, Australia, Europe, and East Asia. The relatively low correlations over Australia and Europe are due to the low climatological mean and variance. While the other low-performance regions are all mixed pollution and dust environment that is challenging for all RAs. Some relatively high-performance regions (total AOD $r^2$ greater than 0.85 for at least two RA members) include Central America, Peninsula Southeast Asia, and Maritime Continent. Total and CM AOD $r^2$ are high over Central America, because it is a receptor region for African dust, and RAs perform well in general during long-range transport over ocean where data assimilation is very effective in correcting model AOD fields. Total and FM AOD $r^2$ are high over Peninsula Southeast Asia, and Maritime Continent, suggesting the RAs can capture the large interannual variabilities of the regional dominant aerosol species, BB smoke, associated with the impact of ENSO on fire activities in the regions (e.g., Reid et al., 2012; Xian, et al., 2013). Overall, the MRC exhibits superior $r^2$ compared to individual RAs for modal AODs regionally and globally.

For remote marine sites, including Ascension Island in the mid-basin of south Atlantic, Ragged Point in the western Tropical Atlantic, Mauna Loa in Hawaii, MCO-Hanimaadhoo in the north Indian Ocean, and REUNION_DENIS in the south Indian Ocean, the RAs exhibit similar performance at these sites as they do over the upwind land or coastal regions (Fig. S1). An exception is Mauna Loa. Mauna Loa is situated at an elevation of 3.4 km, well above the marine boundary layer and remote from continental sources. At this location, all the RAs exhibit a significant positive bias. One possible explanation for this bias is the topographic effect, as the coarse spatial resolutions of the models may not be able to resolve the site's high elevation or its sharp elevation gradient compared to the surroundings. Additionally, uncertainties in the removal processes during long-range transport may also be contributing to the high bias. It is also worth noting that all the RAs do especially well at the Ragged Point site, with total AOD $r^2$ close to or higher than 0.92. This site is a receptor site of African dust in the Western Tropical Atlantic. This suggests that the RAs capture the long range-transport of dust from Africa quite well. This is related to the fact that data assimilation systems have more chance to correct the model fields with observations in the long-range transport over the ocean.

When considering the contribution of dust and sea salt aerosols to FM AOD in CAMSRA, MERRA-2 and JRAero, the verification statistics (bias, RMSE and $r^2$) for the total AOD of these RAs remain unchanged as expected (Fig. S2, S3, S4). However, there is a noticeable shift in the positive bias of FM AOD (and negative bias of CM AOD) for these RAs, particularly in regions influenced by dust, such as North Africa, the Arabian Peninsula, East Asia, Central America, South Asia, and Europe. Specifically, the positive bias in FM AOD becomes more pronounced, and the negative bias in CM AOD becomes more negative in these regions, especially for CAMSRA. It's worth noting that in MERRA-2, there is a change in sign, where the FM AOD bias switches from negative to positive in North Africa and the Arabian Peninsula, while the CM AOD bias changes from positive to negative in these regions. Additionally, the negative FM AOD bias becomes smaller, however the negative CM AOD bias worsens in South Asia within both MERRA-2 and JRAero datasets (Fig. S2). In general, when taking into account the contribution of dust and sea salt aerosols to FM AOD (by default, dust and sea salt AODs are treated as CM AODs in this study) in CAMSRA, MERRA-2, and JRAero, we observe a worsening of both FM and CM AOD biases in these three datasets. Similarly, the RMSE for both FM and CM AODs over regions influenced by dust deteriorates as well (Fig. S3). The $r^2$ for FM and CM AODs in these regions also worsens overall, with the exception of an improvement in FM AOD over Central America. FM sea salt's impact on the verification score is small as the majority of AERONET sites are on land and FM sea salt only contributes on the order of ~10% to total sea salt AOD in the three RAs.

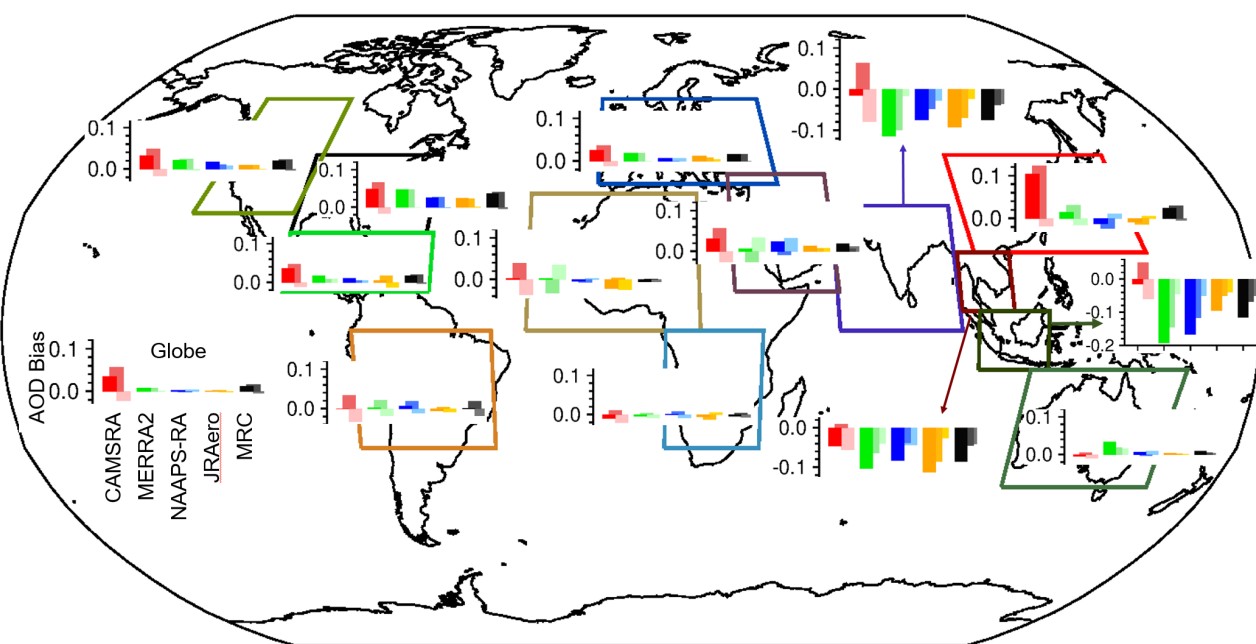

Figure 5. Regional total, FM, and CM AOD biases for the four reanalyses and the MRC compared with AERONET data. Each grouped bars in the same color system present total, FM, and CM AOD biases from left to right (also dark to light).

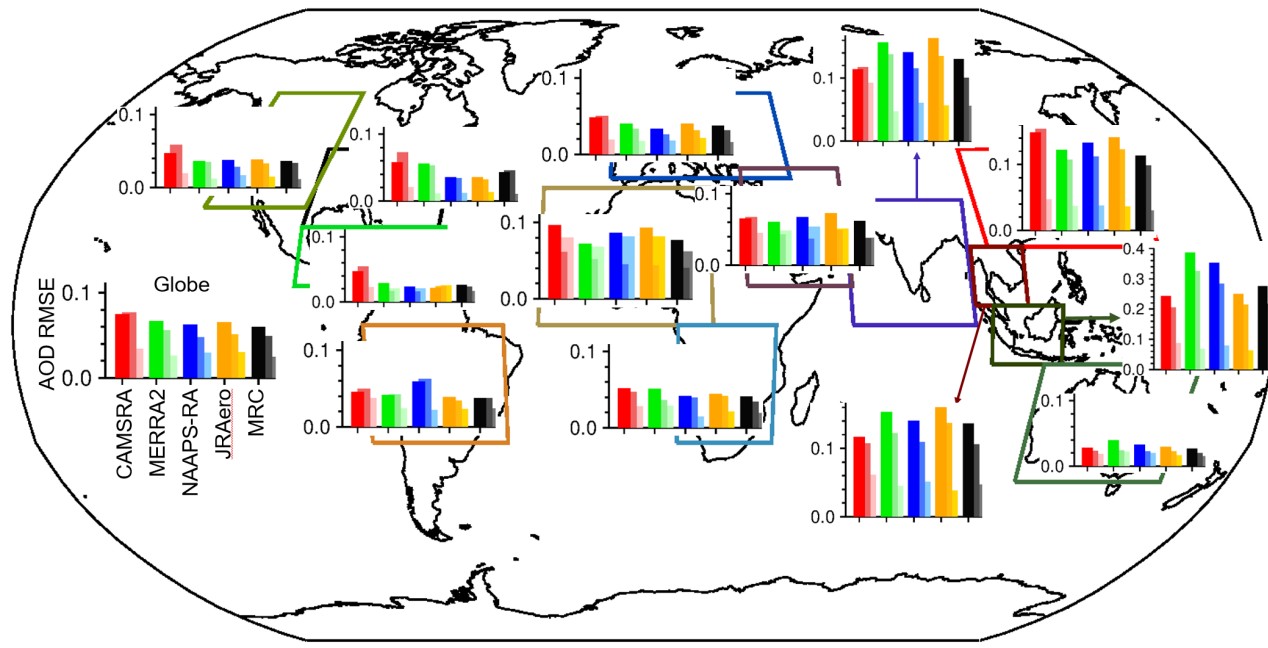

Figure 6. Same as Figure 5, except for AOD RMSE.

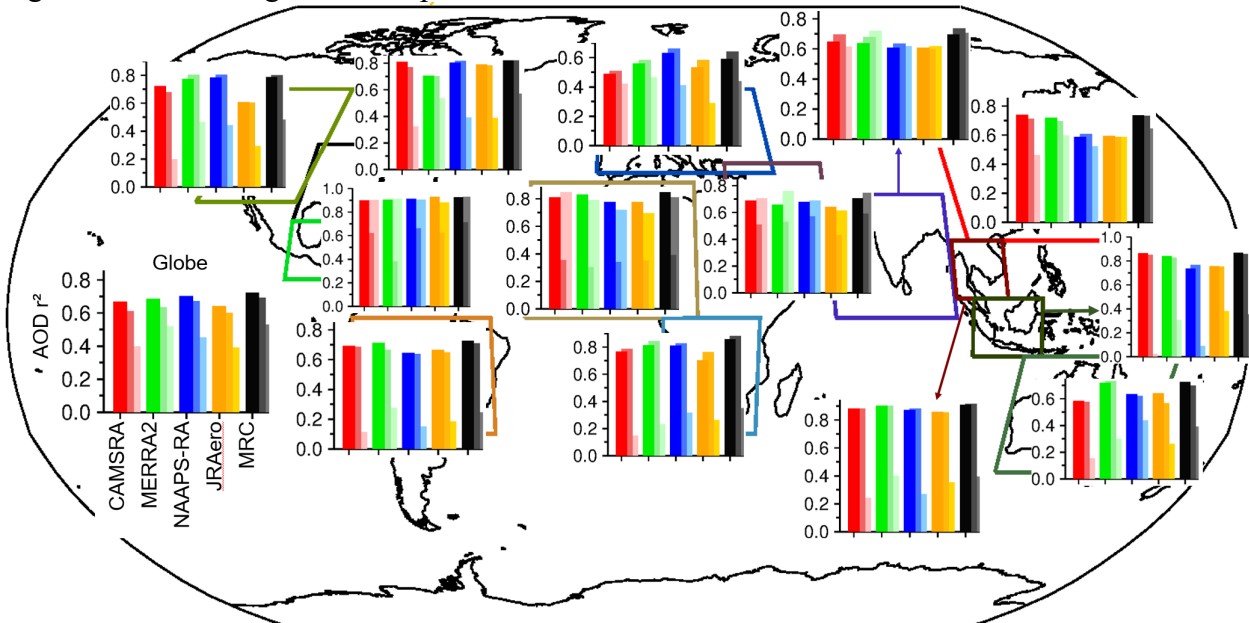

Figure 7. Same as Figure 5, except for the AOD coefficient of determination ($r^2$).
3.3.2 Rankings of the RAs with respect to validation statistics
To expand the validation result from regional averages to individual sites, including remote sites
that are not included in the regional analysis, rankings of the RAs in terms of RMSE of monthly
total AOD at all the AERONET sites are displayed in Figure 8. It shows that there are cases in that
individual RA ranks first over some regions. For example, CAMSRA ranks relatively better than
others in South and Southeast Asia, MERRA2 ranks better over North Africa and Arabia
Peninsula, NAAPS-RA ranks better over North America and Europe while JRAero performs
relatively better over Southern North America and the Caribbean. Individual RA has mixed results
for sites in other regions. AOD RMSE of the MRC is not always the lowest for a given site, but it
is relatively low and stable over the globe. This is consistent with the regional RMSE result (Figure
6). The consensus wins because of its averaging of independent models. This is consistent with
our findings with the ICAP models (Sessions et al., 2015; Xian et al., 2019).
Challenging sites for these RAs are found as marked by the magenta color in Figure 8. These sites
exhibit an $r^2$ value of less than 0.25, and are associated with relatively large AOD bias and/or
RMSE. Often, when a challenge occurs, it is a common challenge to all models, and no specific
model is much better than the others. Some of the causes for the challenges include lack or large
uncertainty in local emissions (e.g. Modena in Northern Italy, Mainz in Germany, Cario_EMA in
Egypt, Trelew and CEILAP-RG sites in Argentina), and/or topographic effects that are not
resolved in these RAs due mostly to coarse model spatial resolutions (e.g., Mauna_Loa), and sites
that are impacted by mixed pollution and dust (Dushanbe in Tajikistan).

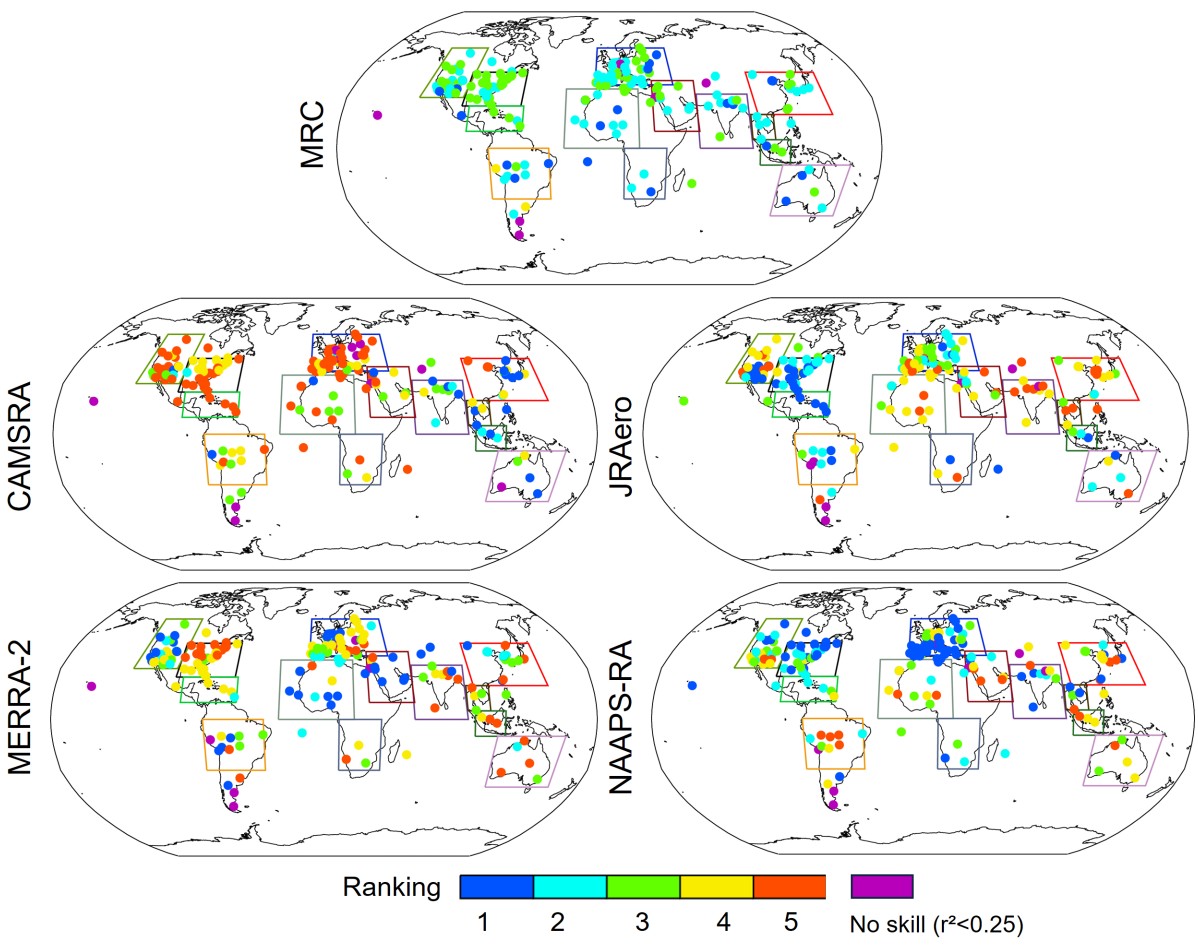

Figure 8: Ranking of aerosol RAs in terms of RMSE of monthly total AOD at 550nm over all the
AERONET sites. Rectangles are used to delineate regions for regional validation, as depicted in
Figures 5, 6, 7. A lower RMSE indicates better performance, with a ranking of 1 being the most
desirable. AERONET sites with a coefficient of determination ($r^2$) less than 0.25 are marked in
magenta, indicating a lack of skill from the model.

Ranking analyses were also conducted on the RMSEs of FM and CM AODs, absolute bias, and $r^2$
of modal AODs. Figure 9 presents the MRC rankings for all these comparison statistics. In line
with the MRC ranking for the total AOD's RMSE, the MRC rankings for other metrics are
predominantly ranked first or second, except for the absolute biases, where MRC rankings are
often ranked third over North America, South Americas, and Europe for total and FM AODs. For
these modes and over these regions, all the RAs have positive biases relative to AERONET. When
the biases are in the same sign (positive or negative), it is mathematically natural for MRC to rank
in the middle. For CM and FM AODs, there are more sites with $r^2$ <0.25 compared to the total
AOD. These sites mostly have small values of CM or FM AODs, and reside in regions of opposite-
mode dominance, such as FM in Saharan region, CM in northern Europe and N. America. From
another perspective, the MRC ranking with respect to correlations is superior to RMSE and then
absolute bias. That is, the MRC better captures aerosol variance than the individual models, but is
nevertheless subject to overall model biases. The MRC ranking for CM AOD is slightly superior
to that of total AOD and then FM AOD. While the MRC ranking is not consistently at the top for
a given site or region, it is relatively high and stable, ranking first for the global average. No
individual RAs could compete with the MRC in that sense.

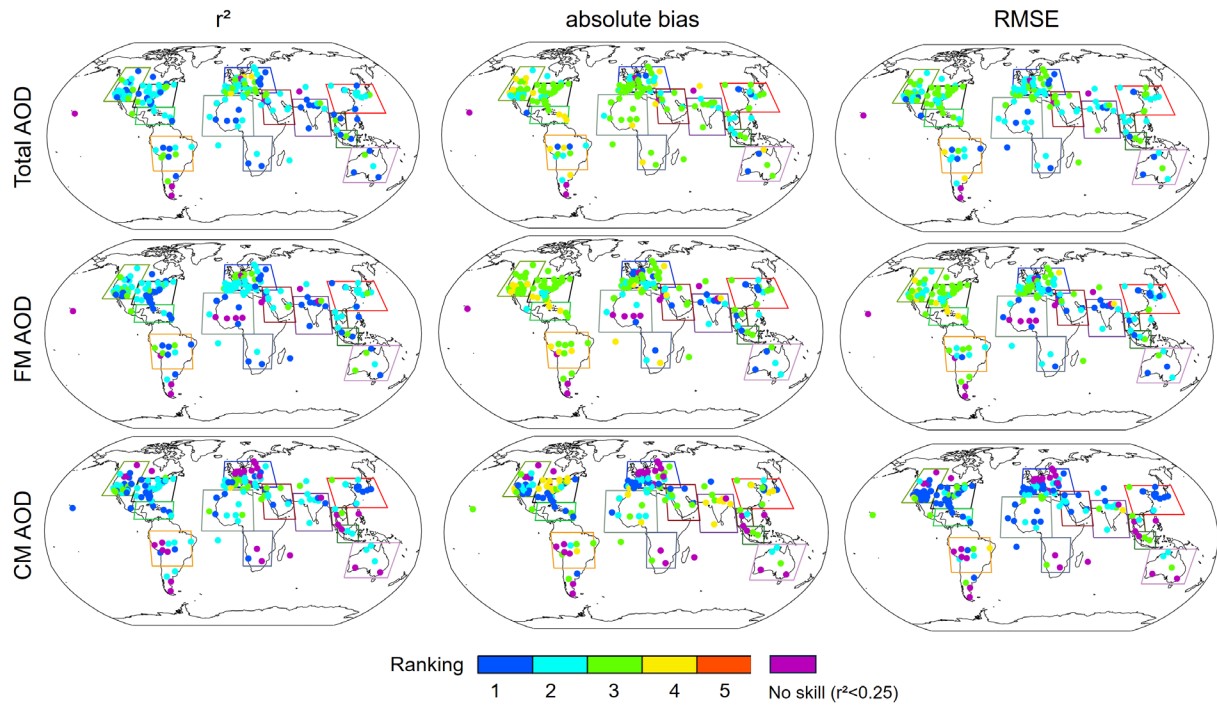

Figure 9. Ranking of the MRC among all the RAs in terms of $r^2$, absolute bias, and RMSE of the
total, FM, and CM AODs over AERONET sites.

3.4 Seasonality of Regional AODs
In Section 3.1 we depict the spatial distribution of total AODs from all the RAs across the four
seasons. In this section, we provide monthly time series of AOD and AOD interannual variabilities
for 16 regions (Fig. 10), along with the contributions of speciated AOD to the total AOD for these
regions for four seasons and the annual-mean. All the RAs exhibit a similar seasonality and

interannual variability of total AOD for all regions, except for the Antarctic and Arctic, particularly during their winter seasons. This disparity arises from the absence of passive satellite AOD data during polar winter, which limits the effect of data assimilation on model AOD (see Xian et al., 2022 for the Arctic region). Even during polar summer, AOD retrievals are often unavailable due to high reflectance from surface ice/snow. The total AOD in JRAero exhibits exceptionally high levels, primarily attributed to elevated sea salt and sulfate AODs (Fig. S5). This anomaly stems from the MASINGAR model used to produce JRAero, which tended to underestimate the removal of aerosols via cumulus convection. Consequently, this led to an overestimation of aerosol concentrations in the polar regions and the upper atmosphere. The underestimation of the removal process has been resolved in the current MASINGAR model and the overestimation of AOD over the polar regions will be improved with the JRAero version upgrade. Nevertheless, the polar regions demonstrate the most significant divergence among the RAs in the seasonal cycle and speciation of AOD.

The regions that are dominated by BB-smoke, including South Africa, South America, Maritime Continent, Peninsula SE Asia, and western North America, exhibit consistent peak seasons of total AOD with their respective burning seasons. The Maritime Continent and Peninsula SE Asia experience extremely large interannual variations of peak monthly AOD, owing to a strong positive correlation between burning activities and El Nino cycles (e.g., Reid et al., 2012; Xian et al., 2013). The contributions of sulfate/ABF AOD induced by pollution are dominant in East Asia and South Asia, while other aerosol species also make a significant contribution to the total AOD. In Europe and East N. America, sulfate/ABF is also the dominant species; however, the monthly total AOD values are much smaller. All the RAs capture the dominance of dust species in summertime over SW Asia and NW Africa. The relatively high AOD in springtime in NW Africa is partially due to BB in Sahel. In Australia, the peak AOD in Oct-Dec is associated with BB smoke. In Central America, the relatively high AOD in the springtime results from BB smoke. Although quite diverse in AOD magnitude, all RAs tend to have a summertime total AOD peak attributed to dust. For the global average, sea salt AOD has a significant contribution to the total AOD as the area of the ocean overwhelms the area of land. Monthly time series of the speciated AODs for all the regions are available in Fig. S5. Overall, the seasonality and interannual variability of total AOD for most regions is very similar among the RAs. Moreover, all RAs have the same dominant species for most regions, but the contributions from different species can be quite different in these RAs. This is a result of the fact that total AOD is constrained within these RAs through data assimilation, while speciated AODs are not. Aerosol speciation and the contribution of each species to the total AOD are determined by the construction of the aerosol forecast models, which are very independent in these RAs.

a)

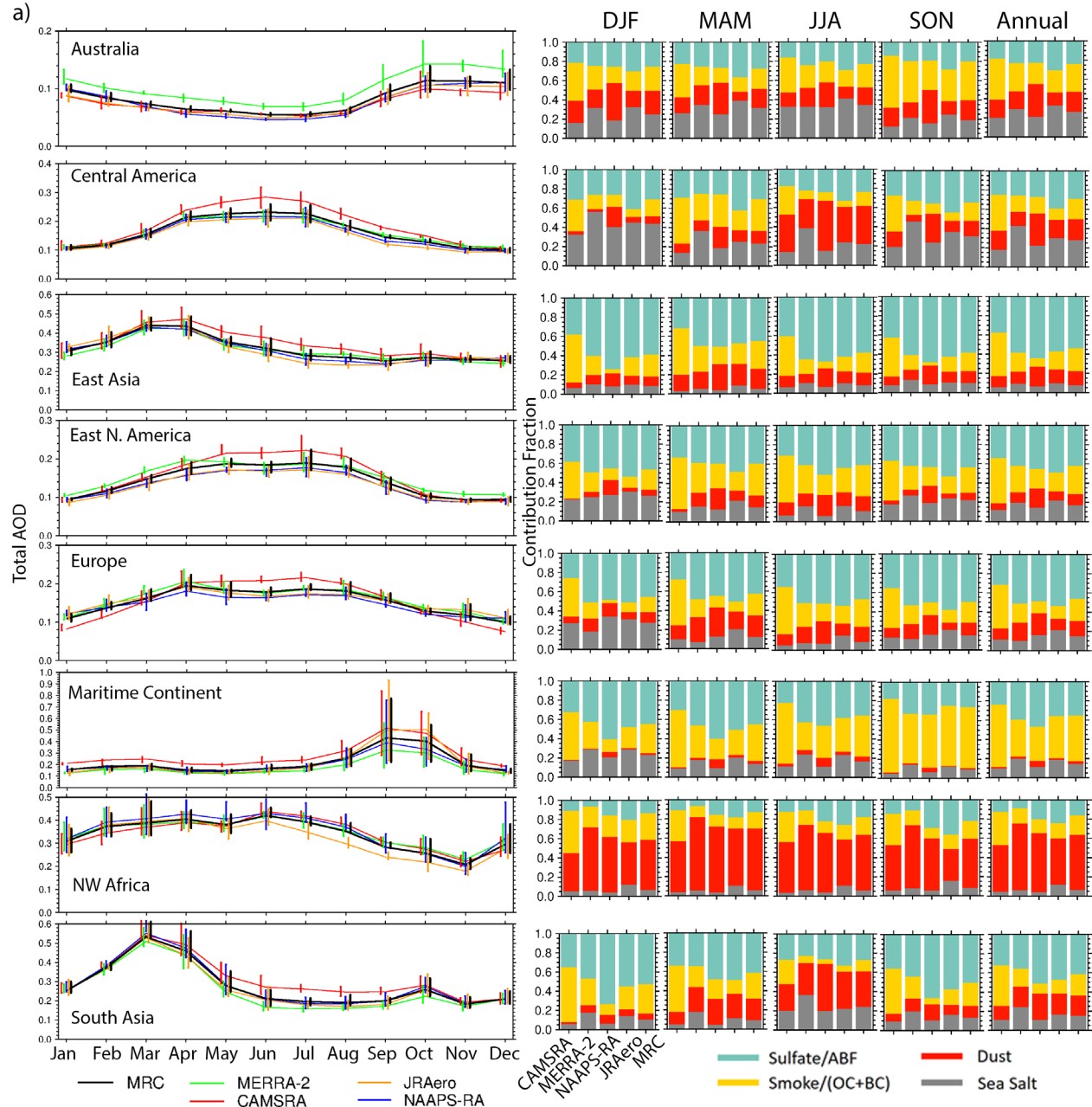

709

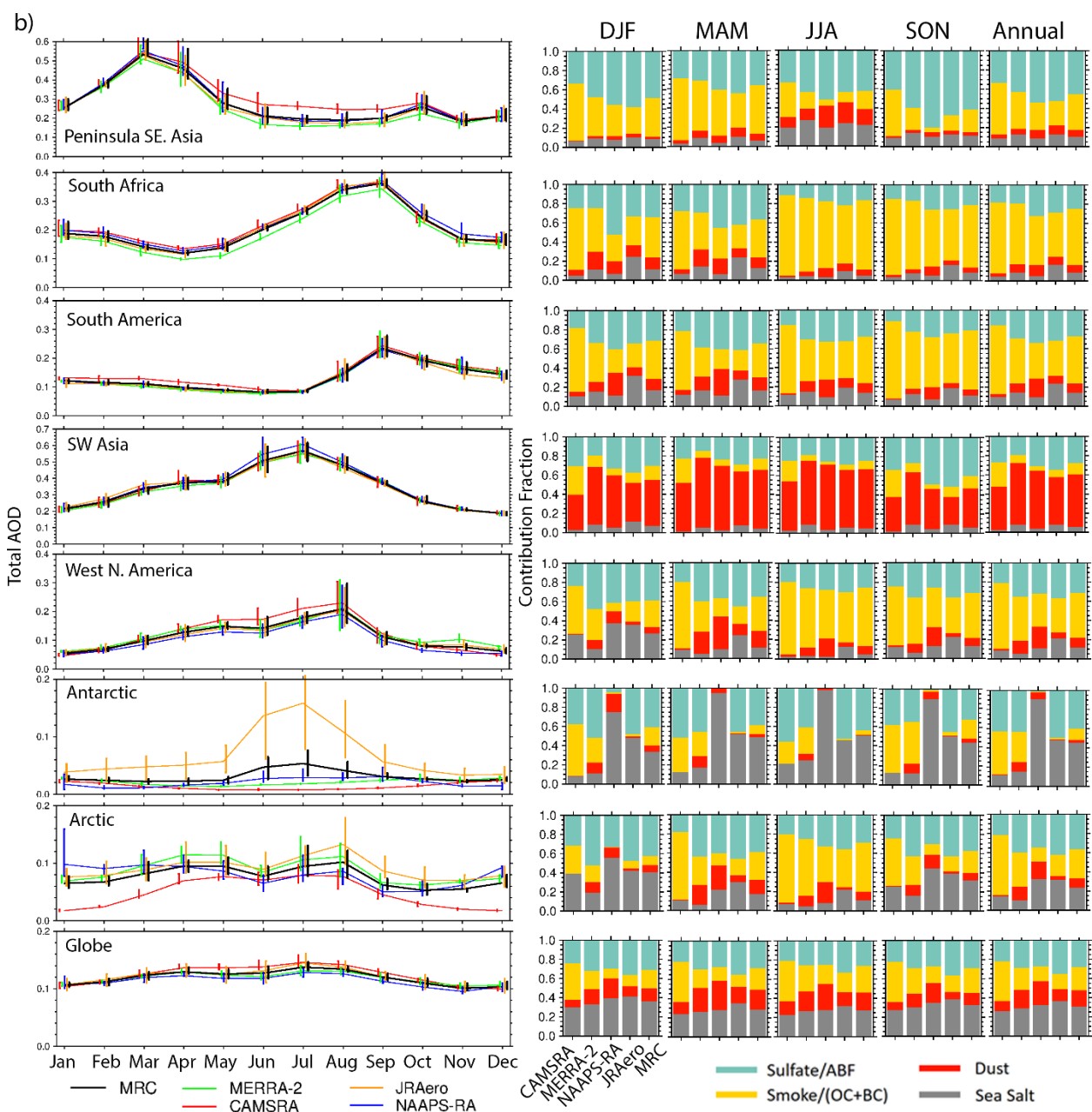

Figure 10. Climatological seasonal cycle of regional mean total AOD (left), and the contribution fraction of speciated AOD to the total AOD for the corresponding regions and seasons from the four RAs and the MRC (right). In the seasonal cycle plots, bars denote the interquartile range of monthly-mean AOD, illustrating interannual variabilities for the period 2011-2019.

3.5 Urban versus Rural areas

To evaluate the RAs for urban versus rural areas, three paired sites were selected. Beijing (China), Yonsei University (South Korea) and Kanpur (India) represent urban areas, while their corresponding rural areas are represented by the Xiang He, Anmyon and Gandhi College sites among the available AERONET sites. Fig. 11 shows the monthly time series of modal AODs from the RAs and the MRC, along with their validation statistics against AERONET data. The dominant aerosol mode is FM at all these sites, due mostly to pollution. These sites are also

subject to the influence of dust storms in springtime, which contributes to CM AOD. The modal
AODs from the four RAs and the MRC generally follow these of AERONET seasonally. The
spread among the RAs is relatively large for the Chinese and Indian sites. The spread is relative
small for the Korean sites, with the spread being slightly less for the rural site Anmyon than for
its corresponding urban site Yonsei University. Regarding bias, RMSE, and $r^2$, there is no
significant difference between the urban and the corresponding rural sites for each RA and the
MRC, despite that $r^2$ of total AOD tending to be higher for the rural sites than the urban sites.
The $r^2$ of FM AOD also tends to be higher than that of the CM. The RAs and the MRC also
capture the decreasing AOD trend in the latter half of the 2011-2019 time period for the Chinese
and Korean sites. A more detailed trend analysis will be provided in a companion paper. For the
ranking of all RAs in terms of bias, RMSE and $r^2$, each individual RA has a few first rankings.
MERRA-2 is especially better compared to other RAs at CM/dust AOD for the Indian sites. But
in terms of the number of ranking first, the MRC is the winner for all the sites (at least having 5
out of 9 statistical variables ranking first for each site).

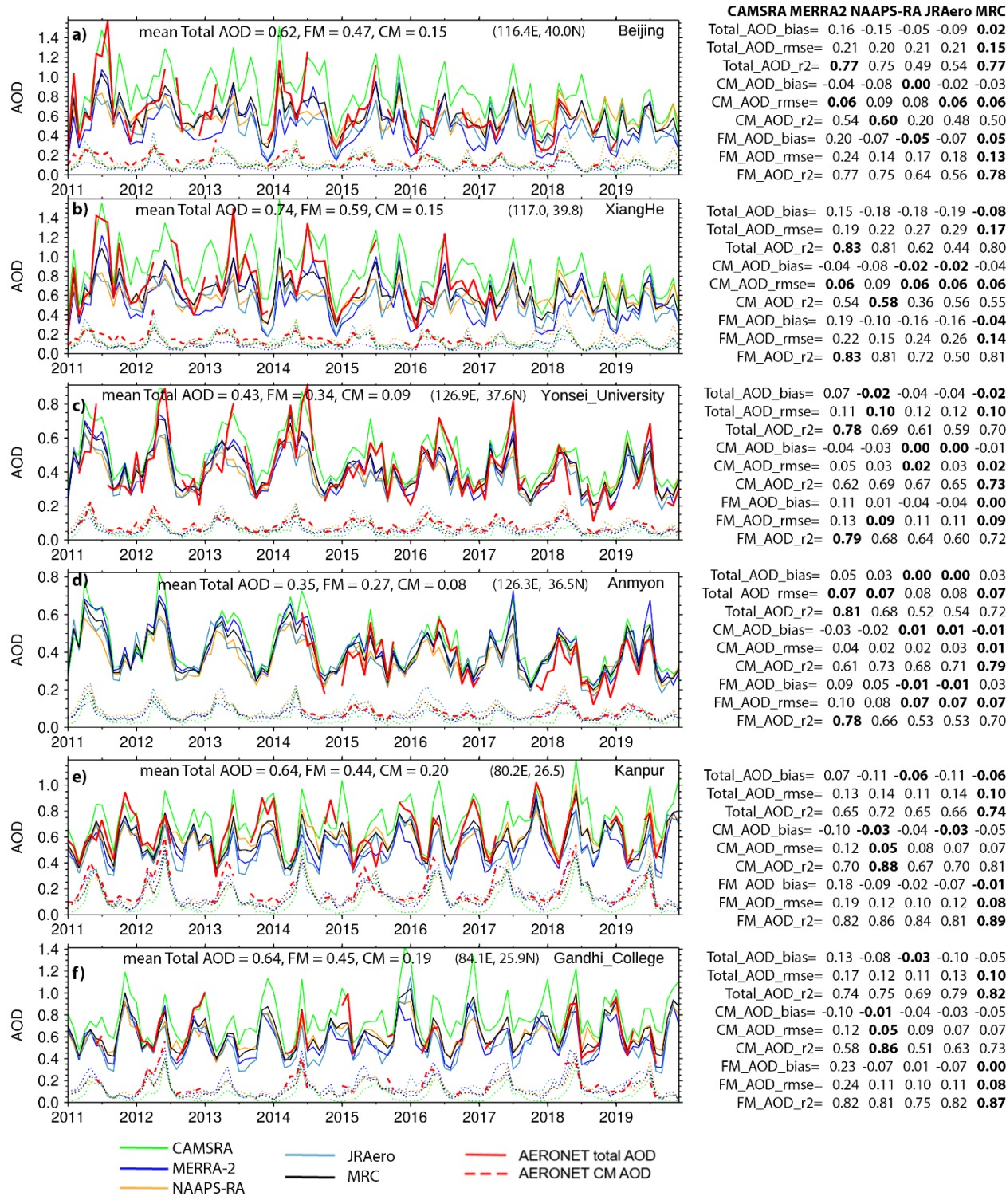

Figure 11. Evaluation of total, FM, and CM monthly AODs from the RAs at urban versus rural AERONET sites. Sites a), c) and e) represent urban locations in China, Korea and India respectively, while sites b), d) and f) denote their corresponding rural sites. Mean Total, FM, and CM AODs from AERONET data are presented in the upper panels of the time series plots for each site. The right column displays verification statistics for the four RAs and the MRC,

including bias, RMSE and r$^2$. Values in bold indicate the lowest bias or RMSE, or the highest r$^2$,
signifying the best ranking among all the RAs.

**4. Conclusions**
This study compares the monthly average total, and speciated aerosol optical depths (AODs) from
four different aerosol reanalyses (RAs). These include the Copernicus Atmosphere Monitoring
Service ReAnalysis (CAMSRA) developed by Copernicus/ECMWF; the Japanese Reanalysis for
Aerosol (JRAero) developed at the Japan Meteorological Agency (JMA); the Modern-Era
Retrospective Analysis for Research and Applications, version 2 (MERRA-2) developed by
NASA; and the Navy Aerosol Analysis and Prediction System reanalysis (NAAPS-RA) version
1, developed by the U.S. Naval Research Laboratory. The consensus of the four RAs is also
developed for intercomparison. The AODs from these RAs are evaluated with AEROsol Robotic
NETwork (AERONET) and the MODIS Dark Target/Deep Blue retrievals (Levy et al., 2013;
Sayer et al. (2014)) using data from 2011-2019. The following are the conclusions drawn from this
study:
1) Global distribution and magnitude of total AOD demonstrate a high level of similarity
among all four RAs. The spread of total AOD among the RAs is small over most regions.
Exceptions, where the RAs diverge in total AOD are polar regions and areas affected by
specific factors that include volcanic outgassing, high terrain, and certain desert regions.
2) The relative spread of speciated AODs is considerably larger than that of total AOD.
CAMSRA consistently yields higher values for biomass burning (BB) smoke or Organic
Matter (OM) AOD in comparison to other RAs. Meanwhile, NAAPS-RA exhibits
generally higher dust AOD values. JRAero has comparatively high biased inland sea salt
AOD. The divergence of speciated AODs in regions remote from aerosol sources is large,
implying different efficiencies in removal during long-range transport. This phenomenon
results from the fact that data assimilation in these RAs constrains total AOD but not
speciated AOD.
3) The seasonality and interannual variability of total AOD in the 16 regions under study,
with the exception of the Antarctic and Arctic, demonstrate a high degree of similarity
across the various RAs and align with the observations. While the dominant species of
aerosols are consistent across most regions in all RAs, the relative contributions from
individual species can vary significantly.
4) The accuracy of the RAs, as measured by RMSE, bias, and correlation of the total, fine-
mode (FM) and coarse-mode (CM) AODs (i.e. modal AODs), has been verified with
AERONET. It is evident that each RA exhibits its own unique regional strengths.
Specifically, CAMSRA performs better in South and Southeast Asia, MERRA-2 excels in
African and Arabian Peninsula dust regions, NAAPS-RA shows relatively better
performance over Europe and East CONUS, and JRAero performs relatively better over
southern North America and the Caribbean. Common challenges to all the RAs often
include lack or large uncertainty in local emissions, and/or topographic effects, as well as

situations where both FM and CM states are mixed. There is no significant difference in RAs' performance for urban versus rural areas, despite that rural areas tend to have slightly higher AOD correlations with observations. RAs show the worst performance in areas impacted by mixed FM and CM aerosols, such as South Asia and East Asia, and areas that experience substantial interannual variability in AOD, for instance, Southeast Asia, and the Maritime Continent. The polar regions present a challenge due to limited observations.

5) The Multi-Reanalysis-Consensus (MRC), an ensemble mean of the four RAs, is not consistently the best performer in terms of RMSE, bias and correlation of modal AODs for a given site or region. However, the MRC generally performs relatively well and remains stable, ranking first or second regionally and first globally among all the RAs, especially for correlation and RMSE. The MRC ranking with respect to correlations is superior to RMSE and then absolute bias. The MRC ranking for CM AOD is slightly superior to that of total AOD and then FM AOD. The MRC method gains an advantage due to its ability to average independent models.

The findings presented in this study offer a comprehensive overview of the current state-of-the-art aerosol RAs in the context of monthly AOD. The strengths and weaknesses of individual RAs and their collective implications will provide valuable information for diverse potential users. Compared to intercomparisons of satellite AOD products, which have shown a typical bias of 15%-25% (which regionally can reach ±50%) and AOD divergence of 10% over ocean to 100% over certain land areas amongst 14 satellite products in Schutgens et al., 2020, and the intercomparisons of different MODIS products shown in Fig. 1, the biases and divergence of AODs from the four RAs are moderate. The MRC product, which is currently a simple ensemble mean of the four RAs, could be potentially improved with regionally-weighted member contributions according to the strengths of the RAs or with aerosol scenario/species-weighted member contributions.

The results of the intercomparison highlight areas for improvement in the next generation of aerosol RAs. These improvements may include tuning of emission sources and sinks, finer spatiotemporal resolutions, incorporation of additional aerosol species, such as nitrate aerosols and dust with different mineralogy, separation of BC and OC from BB emissions in some RAs, and application and enhancement of BB plume rise models. Moreover, some centers are planning to incorporate new observational data, such as OMI Aerosol Index to constrain the amount of absorptive aerosols, which has the potential to enhance simulations of BB smoke and dust aerosols (Zhang et al., 2021; Sorenson et al, 2023). Vertical profiles of aerosol backscatter measured by CALIOP and future space-borne lidars may also be incorporated into RAs to help constrain aerosol vertical distribution. Anticipated advancements in emission inventories, retrieval algorithms, space-borne sensors, upcoming satellite missions, and improvements in meteorological and aerosol modelling are expected to drive progress in aerosol RA.

**Appendix A: Abbreviations:**
ABF: Anthropogenic and Biogenic Fine aerosols
AERONET: Aerosol Robotic Network
AOD: Aerosol Optical Depth
AVHRR: Advanced Very High Resolution Radiometer

BB: Biomass Burning
BC: Black Carbon
CALIOP: Cloud-Aerosol Lidar with Orthogonal Polarization (CALIOP)
CAMSRA: Copernicus Atmosphere Monitoring System Reanalysis
CM: Coarse Mode
FLAMBE: Fire Locating and Modeling of Burning Emissions
FM: Fine Mode
ICAP: International Cooperative for Aerosol Predictions
JRAero: the Japanese Reanalysis for Aerosol
MASINGAR: Model of Aerosol Species IN the Global AtmospheRe
MISR: Multi-angle Imaging SpectroRadiometer
MME: Multi-Model-Ensemble
MODIS: Moderate Resolution Imaging Spectroradiometer
MODIS-DT: MODIS Dark Target
MODIS-DB: MODIS Deep Blue
MODIS-DA: MODIS data assimilation quality data.
MRC: Multi-reanalysis-consensus
NAAPS-RA v1: Naval Aerosol Analysis and Prediction System-Reanalysis version 1.
MERRA-2 :Modern-Era Retrospective Analysis for Research and Applications version 2
OM: Organic Matter
OC: Organic Carbon
OMI: Ozone Monitoring Instrument (OMI)
PMAp: Polar Multi-Sensor Aerosol product
QFED: Quick Fire Emissions Dataset
RA: ReAnalysis
RMSE: Root Mean Square Error
SDA: Spectral Deconvolution Method
**Appendix B: Definition of terminologies**
Root Mean Square Error (RMSE):
$\text{RMSE} = \sqrt{\frac{1}{n}\sum_{i=1}^{n}(\tau_{model} - \tau_{obs})_i^2}$ where $\tau$ represents monthly AOD, and n is the total number
(i.e. month) of observational or model data.
Bias: $\tau_{model} - \tau_{obs}$
Mean error: $\frac{1}{n}\sum_{i=1}^{n}(\tau_{model} - \tau_{obs})_i$
Mean absolute error: $\frac{1}{n}\sum_{i=1}^{n}|\tau_{model} - \tau_{obs}|_i$
Coefficient of determination: $r^2 = \frac{\left(\sum_{i=1}^{n}(x_i-\overline{x})(y_i-\overline{y})\right)^2}{\sum_{i=1}^{n}(x_i-\overline{x})(y_i-\overline{y})\sum_{i=1}^{n}(x_i-\overline{x})(y_i-\overline{y})}$
where $\bar{x}$ and $\bar{y}$ are the mean values of variable $x$ and $y$.
Multi-Reanalysis-Consensus (MRC): $\frac{1}{m}\sum_{i=1}^{m}x_i$ where $m$ is the total number of the individual
reanalysis, which is 4 for this study.
Spread among the RAs is defined as the standard deviation of all the individual models, ie.,
$\sigma = \sqrt{\frac{1}{m}\sum_{i=1}^{m}(x_i - \bar{x})^2}$ where $x_i$ is individual reanalysis, and $\bar{x}$ is the MRC.

### 868 Data Availability

All the data supporting the findings of this manuscript can be accessed via the provided links or
by requesting them using the contact information provided within those links.
AERONET Version 3 Level 2 data: http://aeronet.gsfc.nasa.gov
MODIS data-assimilation-quality AOD:
https://modaps.modaps.eosdis.nasa.gov/services/about/products/c61-nrt/MCDAODHD.html
CAMSRA AOD: https://www.ecmwf.int/en/research/climate-reanalysis/cams-reanalysis
JRAero product: https://www.riam.kyushu-u.ac.jp/taikai/JRAero/
MERRA-2 AOD:
https://disc.gsfc.nasa.gov/datasets/M2TMNXAER_V5.12.4/summary?keywords=%22MERRA-
878 2%22

NAAPS-RA AOD: https://usgodae.org//cgi-
bin/datalist.pl?dset=nrl_naaps_reanalysis&summary=Go
MRC AOD: https://nrlgodae1.nrlmry.navy.mil/cgi-
bin/datalist.pl?dset=nrl_mre4_post&summary=Go

### 884 Supplement


### 886 Author contributions

PX and JSR designed the study. PX performed the data analysis and wrote the paper with
contributions from MA, PRC, KY, TFE, EJH, and JZ on data descriptions and information
collection. All authors contributed to the discussion of the results and revising the paper.

### 890 Competing interests

The contact author has declared that none of the authors has any competing interests.

**Acknowledgments**

The authors acknowledge financial supports from the Office of Naval Research Code 322. Partial support comes from NASA's Interdisciplinary Science (IDS) program (grant no. 80NSSC20K1260). We also thank the NASA AERONET and MODIS teams for the AOD data used in the study. We extend our gratitude to NASA GMAO, ECMWF, JMA, U.S. ONR and NRL for providing access to the aerosol reanalysis products.

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
