# Peer review of "Intercomparison of Aerosol Optical Depths from four reanalyses and their multi-reanalysis-consensus"

_EGUsphere, 2023_

## Referee Comment (RC1)

**Intercomparison of Aerosol optical Depth from four reanalysis and their multi-reanalysis consensus**                                                    by J. Peng et al.

**Strengths**
The consensus is a useful product to study regional AOD trends over the last decades
Insightful presentation of host model biases in data assimilations
map presentations to illustrate global distributions of component AOD

**Weaknesses**
The component separation depends on host model input and process assumptions
A model's component skill may suffer from total AOD bias corrections in assimilations
Assimilated satellite AOD has built/in uncertainties (absorption, albedo, cloud-clearing)
The (coarse) monthly temporal scale of the consensus model limits it application.

**General**
        The paper investigates monthly average (mid-visible) AOD maps of MODIS retrieval assimilations of 4 different models and of their average (the consensus) over the last two decades. In addition, the total AOD values are stratified both into maps for sub-micrometer (fine-mode) and super-micrometer (coarse-mode) particles and into maps of contributing aerosol components (SU, OC, BC, DU, SS). Hereby the component separation, if components are spectrally defined (not only by size but also absorption) opens the door for aerosol associated climate impact assessments.
        Even though all models rely primarily on the same near real time MODIS AOD retrievals in their assimilations, resulting AOD maps differ, despite overall similar locations of AOD maxima. AOD stratifications into aerosol size modes and even in more detailed aerosol component AOD maps illustrate that these differences are driven by much larger diversity in forecast (emission input, aerosol processing) models. In howfar any component separation skill of modeling though is compromised by AOD bias corrections in the assimilation process remains unclear, To offer a more stable data product for AOD trend investigations over the last two decades and at the same time also to illustrate tendencies of individual models, the concept of an average mode (the consensus, MRC – a reduced ICAP version?) was developed. The consensus AOD values appear realistic (also compared to an alternate top-down approach). Only fine-mode AOD and organic contributions (if absorbing in the mid-visible) are stronger.
        Thus, the consensus (with its more moderate behavior) is realistic and with its monthly resolution quite interesting for AOD trend analysis (and apparently there is such companion paper), Considering though that assimilations of satellite data add spatial context, the development of a consensus with daily temporal could be very useful to assist aerosol IOPs or even calculations of radiative surface energy budgets.
        For climate impacts (at TOA) however, aside from size also aerosol absorption increasingly matters, so that for such an application aerosol components needs to be associated with characteristic spectral absorption behavior.

Overall the paper is quite informative.

**Details**

general    The NRL group (of the main authors) had offered an ICAP model in the past based on seven models and it would be interesting how different the new MRC consensus is for common years.

general    The split in components via maps is a nice illustration over which regions and which seasons particular components are important. But to go the next step (to address aerosol radiative effects) each of these components need to be associated with information of size (distribution) and composition (spectral refractive indices – also addressing absorption), so that needed single scattering properties for radiative transfer can be offered.

general    Consensus ???  would not be a '4-model average' be more precise? (An alternate method would be to exclude the largest and smallest model … but then components would not be additive anymore … so … not a good choice here)

Line 64    Why is the consensus better than ICAP? Because each of the 4 data-set addresses all (5 aerosol) components (to add up totals)? In the end (when presenting data) it also would be interesting how total, FM and CM AOD would differ between ICAP and MRC.

Line 83    The use of additional satellite remote sensing data beyond MODIS has probably only a small impact, as this additional volume is small compared to MODIS

Line 117/231 These are nice model descriptions (very useful) but they could be a bit extended. However, since aerosol components are a major element of the model inter-comparison, the model comparisons could be more insightful with a comparison of aerosol microphysical assumptions for each component (possibly in a table), as size and absorption (of each component) are needed for the transition from aerosol (dry-) mass into aerosol optical depth (e.g. reff of log/normal size/distributions or bin/schemes, mid-visible imaginary part of the refractive index) and mass-extinction efficiency (component AOD/ component_dry_mass) also informs on the assumed aerosol water uptake.

Line 253/257 The SDA method assumes 0.5um radius split, while the split using the 22 size bins of the AERONET inversion is at a radius at 0.528um. This is very close. And since a bi-modal distribution has usually a minimum for these sizes the differences in the AOD splits (into FM and CM) are likely small. The small high bias by the SDA may also be possibly associated with its smaller 500nm (compared to the 550nm) reference wavelength.

Line 260      I would have removed high mountain sites for evaluation (e.g. Mona Loa, Izana, …) to begin with - as it will introduce biases in regional (100km2) evaluations

Line 272      What about the AOD overestimates at low AOD by MODIS? Just compare to other retrievals over oceans (e.g. MISR, SeaWifs…)

Line 288      Why not using the few MAN data at least for polar summers and why not including MAN data in general (giving also a FM/CM split reference over oceans)?

Line 319      Figure 1: nicely chosen color-scale to document differences in the most frequent 0.04 to 0.2 AOD range. I wonder though about the MODIS data, which seem to be rather low. If this is the NRL cleaned version, I would also show the standard MODIS version with higher oceanic AOD, because those data are used in non-NAAPS assimilations. This also would explain higher oceanic AODs for those other assimilations.

Line 336      At higher latitudes the lack of sun-light is contributing factor but the main reason for no data over continents then is the (bright) snow cover.

Line 346      It is not always so obvious that clear-sky AOD is always smaller than all-sky AOD - as stated. In some models it is just the opposite, when wet-removal effects exceed potential aerosol swelling effects at higher ambient relative humidity.

Line 347      Figure 2 … I suggest to use the same color scale for absolute MRC data in column 1 as in Figure 1. And the color scale for differences should also be changed to indicated larger and smaller values (right now only larger values are well indicated … e.g. a green color if values are similar, blue scaling for negative and red scaling for positive …?) The NAAPS ABF might be better split into scattering and carbon components … maybe with a scaling from participating other models. Otherwise the consensus SU component is biased high and OC is biased low. Also, the JRA dust is very low and it likely biases the dust consensus low. Otherwise (if you do not care that components add up to total thus not really recommended) you could also remove unfavorable versions for specific components of the consensus.

Line 369      Emissions are the more likely reason, because removal/transport would also show a similar behavior for sulfate, which is not observed.

Line 430      Figure 3: ratios are nice (possibly also DU/CM and SU/FM) the JRA SU is very large and even larger than the NAAPS SU/ABF.

Line 437      The components of the MACv3 climatology (own work) derived from absorption associated FM AOD and CM AOD (along with assumed component properties) yields similar global component mid-visible AOD averages (for annual

distributions see the maps below): CM 0.060, FM 0.059, DU 0.025, SS 0.035, SU 0.037, OC 0.017, BC 0.006. The consensus has a larger fine-mode contribution and here a larger OC fraction of about 0.020 globally. I would be interested to look at map differences between the consensus and MACv3. MACv3 monthly 1x1 are mid-visible AOD data (same wavelength, same resolution) are accessible (in netcdf) on anonymous ftp in directory

ftp-projects.mpimet.mpg.de/aerocom/climatology/MACv3_2022/550nm_bands20
in file …   MACv3_550nm.nc

Line 457       This is an important point and also a reason why component detail has larger uncertainties than AOD combined totals.

Line 511       I suggest to remove (AERONET) mountain sites and do not go fishing for unlikely (at best secondary) explanations

Line 543       Figure 5: it is different to see the coarse-mode biases. If you single out a different shade (here lighter color for CM) then use to for total and give FM and CM the same shade.

               Figures 6/7: I need more help with the RMS definition. They probably involve data from the same model for all 20 years and also 12 months?  It would be more meaningful to have the average seasonality removed, at least for regions with stronger seasonality … Anyway, I would move figure 6 and 7 in the supplement.

Line 631       It might be nice to show regional component mixtures not annually but for all four seasons

Line 661       This point certainty important. A total bias correction can worsen a (more) skillful component distribution of the forecast model. Even though the maps are illustrative and instructive, these component distributions are not free of error. This is also the reason, why skill I best tested for total AOD.

Line 719       It is nice to offer data access to the four assimilations, but why is a web-location of MRC data missing?

[Figure]

(ann) MAC

AOD,550nm components

---

## Referee Comment (RC2)

**RC Reviewer Comment 2 on egusphere-2023-2354**

Peng Xian et al.:

**Intercomparison of Aerosol Optical Depths from four reanalyses and their multi-reanalysis-consensus**

This paper is comparing and analyzing the difference of aerosol reanalyzes (RA) from NAAPS-RA, JRAero, NASA MERRA-2, CAMSRA and a consensus of these for RAs named multi-analyisis consensus (MRC). The parameter used for the comparisons are the AOD (Aeorosol Optical Depth), and the FM (fine mode) part of the AOD, ad the CM (Coarse mode) part of the AOD. The four RAs are compared to each other and the RMC is confronted to all four methods also. As evaluation, AODs from these RAs are compared with AEROsol Robotic NETwork (AERONET) and with the combined MODIS Dark Target/Deep Blue retrievals.

The paper is well written, in perfect English, al the parts are very clear.

The paper is brilliant in showing the differences between the four RAs and the RMC, also in showing, commenting and analyzing the trends and seasonal and geographical variabilities of AOD, FM-AOD an CM-AOD.

The paper suffers of a lack of an explanation concerning the methods. Especially in introduction and part 2. We would expect a longer explanation about what is meant with the "consensus" (RMC). The RMC should be better explained in details: Is it an average of all four other methods? How it differs to a simple average, are there situations/regions/seasons/periods for which one method will have a larger or a weaker weight in the consensus compared to others? If yes why? -> A real effort should be done for explaining better the methods and specificities of this RMC "consensus". Also, you should explain what are the needs of this consensus. Are the four RAs methods not enough?

Other lack of explanation in the methods is for the method of comparison AERONET vs. RAs and RMC. The comparison method should here be much better explained (Part 3.3). Especially for the regional comparison: Which stations are taken account for a given region? Is the AERONET value used in the comparison for the region an average of all stations present in the region and accepted for the comparison regarding the criteria explained in Part 2.5? If yes, is it a relevant method = are all n AERONET sites of the region representative with the same weight 1/n for this region? How do you deal then with sub-regions with higher concentrations of AERONET stations? Are there then not over-weighted in the computation of the "regional AERONET-AOD"? Do you have considerations concerning urban and rural AERONET sites?

These for the general scientific comments.

A general presentation comment is that they are a lot of acronyms used. Positive is that the authors very often detailed in the text several times the meaning of the acronym. Nevertheless, I suggest you to add an acronym table in order to help the reader to understand quickly the acronyms.

I have some specific comments/questions for two pictures:

- Maps of Figure 1: First line of maps (MODIS): I understand that on the Antarctic region during JJA there are no measurements, but why is it the same during the other seasons (example austral summer in DJF)?

- Figure 10: 10b) On the Antarctic Graphic: It is so obvious that JRAero is much more overestimating the AOD that it is worth that you give an explanation in a comment in the text of the article

And a comment about References:

- The most important cited reference (Sessions et al. 2015), cited L104, L235, L292, L351 is missing in the "References" part.

- Sessions et al. 2016 (cited in L66-67) is missing in the References list

---

## Author Comment (AC1)

**Reply to the review comments from Dr. Kinne on "Intercomparison of Aerosol Optical Depths from four reanalyses and their multi-reanalysis-consensus"**

Review comments are in italic font, and our replies are in regular font.

*Strengths*
*The consensus is a useful product to study regional AOD trends over the last decades. Insightful presentation of host model biases in data assimilations map presentations to illustrate global distributions of component AOD.*
*Weaknesses*
*The component separation depends on host model input and process assumptions*
*A model's component skill may suffer from total AOD bias corrections in assimilations*
*Assimilated satellite AOD has built/in uncertainties (absorption, albedo, cloud-clearing)*
*The (coarse) monthly temporal scale of the consensus model limits it application.*
*General*
*The paper investigates monthly average (mid-visible) AOD maps of MODIS retrieval assimilations of 4 different models and of their average (the consensus) over the last two decades. In addition, the total AOD values are stratified both into maps for sub-micrometer (fine-mode) and super-micrometer (coarse-mode) particles and into maps of contributing aerosol components (SU, OC, BC, DU, SS). Hereby the component separation, if components are spectrally defined (not only by size but also absorption) opens the door for aerosol associated climate impact assessments.*
*Even though all models rely primarily on the same near real time MODIS AOD retrievals in their assimilations, resulting AOD maps differ, despite overall similar locations of AOD maxima. AOD stratifications into aerosol size modes and even in more detailed aerosol component AOD maps illustrate that these differences are driven by much larger diversity in forecast (emission input, aerosol processing) models. In how far any component separation skill of modeling though is compromised by AOD bias corrections in the assimilation process remains unclear, To offer a more stable data product for AOD trend investigations over the last two decades and at the same time also to illustrate tendencies of individual models, the concept of an average mode (the consensus, MRC – a reduced ICAP version?) was developed. The consensus AOD values appear realistic (also compared to an alternate top-down approach). Only finemode AOD and organic contributions (if absorbing in the mid-visible) are stronger. Thus, the consensus (with its more moderate behavior) is realistic and with its monthly resolution quite interesting for AOD trend analysis (and apparently there is such companion paper), Considering though that assimilations of satellite data add spatial context, the development of a consensus with daily temporal could be very useful to assist aerosol IOPs or even calculations of radiative surface energy budgets. For climate impacts (at TOA) however, aside from size also aerosol absorption increasingly matters, so that for such an application aerosol components needs to be associated with characteristic spectral absorption behavior.*

Reply: The authors appreciate the reviewer's comprehensive comments. We have followed the constructive comments and revised the manuscript. Our detailed responses are below.

*General The NRL group (of the main authors) had offered an ICAP model in the past based on seven models and it would be interesting how different the new MRC consensus is for common years.*

Reply: It is worth clarifying that the study here is focused on the aerosol reanalyses (RA), which were generated with a fixed aerosol model version for each RA, while the ICAP models are operational models and they receive upgrades from time to time. The ICAP models include the operational models from the four centers (4-core) which also generated their RAs. The model version that was used to generate the RA may or may not be the same as its most recent operational version. However because the AOD retrievals these RAs assimilated (e.g. MODIS C6 AOD) don't change much in recent years, we expect the MRC differ little from the ICAP 4-core model ensemble mean at their analysis time for regions with abundant observational data. The performance of DA models vs. Non-DA models, and the performance of DA models at their analysis time vs. their forecast time do differ as shown in Xian et al. (2019; Figure 9 shows global distributions of yearly average ICAP ensemble mean and ensemble spread of dust AOD from the DA models at their analysis mode and the differences from their forecast mode, and the differences from all models at the analysis and forecast modes).

*general The split in components via maps is a nice illustration over which regions and which seasons particular components are important. But to go the next step (to address aerosol radiative effects) each of these components need to be associated with information of size (distribution) and composition (spectral refractive indices – also addressing absorption), so that needed single scattering properties for radiative transfer can be offered.*

Reply: We've now supplied a new table (new Table 2) with the parameters representing microphysical (size bins) and optical properties (extinction efficiency, single scattering albedo) for each species from the reanalyses.  Another new table, table 3, is provided to show the impact of aerosol water uptake on optical properties, to facilitate potential radiative transfer calculations.

*general Consensus ??? would not be a '4-model average' be more precise? (An alternate method would be to exclude the largest and smallest model … but then components would not be additive anymore … so … not a good choice here)*

Reply: By consensus, we mean mathematical mean. The "consensus" is better and explicitly explained in the introduction, abstract and especially the data/method sections to avoid confusion. "Consensus" is the same as "4-model average". Excluding the largest and the smallest model values would result in two-model average, and cut the participating model numbers to half. From the ICAP experience, to be inclusive in the building of multi-model ensemble is beneficial, despite that some models have larger biases compared to the rest of the models (Xian et al., 2019).

*Line 64 Why is the consensus better than ICAP? Because each of the 4 data-set addresses all (5 aerosol) components (to add up totals)? In the end (when presenting*

*data) it also would be interesting how total, FM and CM AOD would differ between ICAP and MRC.*

Reply: We didn't claim the MRC is better than ICAP multi-model-ensemble (MME). MRC and ICAP are different products and their major difference lies in consistency in time. MRC is based on aerosol reanalyses that are generated with the same aerosol model and with as much as possible consistent observational constraints. So the MRC's performance are expected to not change with time as we will show in the companion trend paper. However, ICAP MME is an operational product, and its participating models experience updates from time to time. For example, many of the ICAP models did a few updates in the past ten years. Some of the updates are major, including addition of more aerosol species, and starting assimilating AOD, increased model resolution, updated parametrization of physical, chemical and/or optical properties and processes. These updates significantly impacted individual model's performance, therefore the performance of ICAP MME evolved with time. "ICAP-MME performance in terms of modal AOD RMSEs of the investigated 21 regional representative sites over 2012–2017 shows a general tendency for model improvements in fine-mode AOD, especially over Asia." from the conclusion of Xian et al.2019.

*Line 83 The use of additional satellite remote sensing data beyond MODIS has probably only a small impact, as this additional volume is small compared to MODIS*

Reply: We agree. This information is now included in the text.

*Line 117/231 These are nice model descriptions (very useful) but they could be a bit extended. However, since aerosol components are a major element of the model intercomparison, the model comparisons could be more insightful with a comparison of aerosol microphysical assumptions for each component (possibly in a table), as size and absorption (of each component) are needed for the transition from aerosol (dry-) mass into aerosol optical depth (e.g. reff of log/normal size/distributions or bin/schemes, mid-visible imaginary part of the refractive index) and mass-extinction efficiency (component AOD/ component_dry_mass) also informs on the assumed aerosol water uptake.*

Reply: To address this comment and the reviewer's earlier comment, we've supplied two new tables. Table 2 shows the microphysical and optical properties for the aerosol species. Table 3 shows the impact of aerosol water uptake on optical properties for the hydrophilic aerosol species for the four reanalyses.

*Line 253/257 The SDA method assumes 0.5um radius split, while the split using the 22 size bins of the AERONET inversion is at a radius at 0.528um. This is very close. And since a bi-modal distribution has usually a minimum for these sizes the differences in the AOD splits (into FM and CM) are likely small. The small high bias by the SDA may also be possibly associated with its smaller 500nm (compared to the 550nm) reference wavelength.*

Reply: Yes, indeed. We included O'Neill et al., 2023 reference about the difference between the SDA and AERONET inversion (or sub-micro fraction) methods for more details for interested readers.

*Line 260 I would have removed high mountain sites for evaluation (e.g. Mona Loa, Izana, …) to begin with - as it will introduce biases in regional (100km2) evaluations*

Reply: Thank you for the suggestion. However we intend to be inclusive and comprehensive about what the reanalyses can do and cannot. So we have kept these sites in our analysis. By the way, Mauna Loa is not included any of the studied regions. It is singled out as an example showing the RAs perform worse under the elevated mountainous scenario. The inclusion of other high mountainous sites in regional verifications affect little of the regional verification statistics, due to their small weight in the regional verification (total number of sites are far more than the number of mountainous site). Also not all mountainous sites see bad skill in the reanalyses. These reanalyses tend to do OK on plateau, but not well for mountain sites that have large elevation gradient with the surroundings.

*Line 272 What about the AOD overestimates at low AOD by MODIS? Just compare to other retrievals over oceans (e.g. MISR, SeaWifs…)*

Reply: We are aware of the AOD overestimate at low AOD by MODIS. Responding to your other comments, we have now included two more MODIS products (standard MODIS-DT and MODIS-DB AODs) in addition to the MODIS-data-assimilation quality data. Figure 1 and the result section is also correspondingly updated.

*Line 288 Why not using the few MAN data at least for polar summers and why not including MAN data in general (giving also a FM/CM split reference over oceans)?*
Reply: This study is focused on monthly-mean AOD evaluation. MAN data is so sporadic over the ocean, it cannot form monthly data, thus not being used. However MAN data was used to evaluate three of the reanalyses focusing on the Arctic region: (Xian et al., 2022 https://acp.copernicus.org/articles/22/9915/2022/)  and over the ocean for the ICAP-core four models (Reid et al, 2022 https://www.mdpi.com/2072-4292/14/13/2978)

*Line 319 Figure 1: nicely chosen color-scale to document differences in the most frequent 0.04 to 0.2 AOD range. I wonder though about the MODIS data, which seem to be rather low. If this is the NRL cleaned version, I would also show the standard MODIS version with higher oceanic AOD, because those data are used in non-NAAPS assimilations. This also would explain higher oceanic AODs for those other assimilations.*

Reply: We have now included the standard MODIS-DT and MODIS-DB AODs in Figure 1 and the associated result description in addition to the MODIS-data-assimilation-quality (MODIS-DA) AOD data and result. The MODIS-DA AOD is bias-corrected so it has a slightly lower AOD (on the order of 0.02) over ocean compared to the standard-

MODIS-DT product. The 0.02 slight high bias from the standard MODIS-DT product can also be seen from Table 7 from (Reid et al., 2022 https://www.mdpi.com/2072-4292/14/13/2978 )

*Line 336 At higher latitudes the lack of sun-light is contributing factor but the main reason for no data over continents then is the (bright) snow cover.*
Reply: Thanks. "The high snow/ice coverage" factor is now added in the sentence.

*Line 346 It is not always so obvious that clear-sky AOD is always smaller than allsky AOD - as stated. In some models it is just the opposite, when wet-removal effects exceed potential aerosol swelling effects at higher ambient relative humidity.*
Reply: After adding the standard MODIS-DT and MODIS-DB AOD in our analysis, the result is rewritten as mentioned earlier. This sentence is removed to avoid confusion.

*Line 347 Figure 2 … I suggest to use the same color scale for absolute MRC data in column 1 as in Figure 1. And the color scale for differences should also be changed to indicated larger and smaller values (right now only larger values are well indicated … e.g. a green color if values are similar, blue scaling for negative and red scaling for positive …?) The NAAPS ABF might be better split into scattering and carbon components … maybe with a scaling from participating other models. Otherwise the consensus SU component is biased high and OC is biased low. Also, the JRA dust is very low and it likely biases the dust consensus low. Otherwise (if you do not care that components add up to total thus not really recommended) you could also remove unfavorable versions for specific components of the consensus.*

Reply: Thanks for the suggestion for better visualization. We've updated the color scale for MRC data in column 1 as suggested. The color scale for AOD differences are also updated, so that blue colors represent negative differences and warm colors represent positive differences, and the color for AOD difference falling within [-0.01, 0.01] is white. For aforementioned reasons, we have kept all four RAs in the MRC.

*Line 369 Emissions are the more likely reason, because removal/transport would also show a similar behavior for sulfate, which is not observed.*
Reply: Your observation is right. We have removed "the less efficient removal". Secondary production of OM is also possible. So the sentence is now revised to "……CAMSRA may have higher BB emissions and/or higher secondary production of OM compared to the other RAs".

*Line 430 Figure 3: ratios are nice (possibly also DU/CM and SU/FM) the JRA SU is very large and even larger than the NAAPS SU/ABF.*
Reply: Yes. The sentence is revised as "The contribution of sulfate/ABF AOD to total AOD ranges from 23% to 34%, with the highest contribution observed in JRAero, *even larger than the ABF AOD contribution in NAAPS-RA*" where the italic part is the new addition.

*Line 437 The components of the MACv3 climatology (own work) derived from*

*absorption associated FM AOD and CM AOD (along with assumed component properties) yields similar global component mid-visible AOD averages (for annual distributions see the maps below): CM 0.060, FM 0.059, DU 0.025, SS 0.035, SU 0.037, OC 0.017, BC 0.006. The consensus has a larger fine-mode contribution and here a larger OC fraction of about 0.020 globally. I would be interested to look at map differences between the consensus and MACv3. MACv3 monthly 1x1 are mid-visible AOD data (same wavelength, same resolution) are accessible (in netcdf) on anonymous ftp in directory ftp-projects.mpimet.mpg.de/aerocom/climatology/MACv3_2022/550nm_bands20 in file … MACv3_550nm.nc*

Reply: The relatively high fine-mode fraction in the MRC is largely due to the contribution from the CAMSRA, which has larger FMF than the other three RAs. We've downloaded the MAC climatology data and compared the MRC climatology with it. Below are results for annual-mean AOD difference between MAC and MRC (MAC-MRC) for different species. In general, MAC has higher dust AOD over Africa and India, higher sea salt AOD over mid-high latitudes, higher BC and sulfate over all, but MAC OC/OA is smaller than that from MRC (CAMSRA has high-biased OA though).

[Figure]

Total 550nm AOD difference: MAC-MRC

[Figure]

[Figure]

Dust AOD difference: MAC-MRC

[Figure]

[Figure]

OM difference: MAC-MRC, using record 8 of the aer_data_ann variable in the MAC netcdf file. Note that OM and OC are defined differently.

[Figure]

Sea salt AOD difference: MAC-MRC

[Figure]

BC AOD difference: MAC-MRC

[Figure]

Sulfate/ABF AOD difference: MAC-MRC

Line 457 This is an important point and also a reason why component detail has larger uncertainties than AOD combined totals.
*Reply: True.*

*Line 511 I suggest to remove (AERONET) mountain sites and do not go fishing for unlikely (at best secondary) explanations*
Reply: We intend to be inclusive and comprehensive about what the reanalyses can do well and what they cannot. Mauna Loa site is an illustration of what RAs don't do well. For the 200 selected AERONET sites, only 5 sites have elevation exceeding 3 km, and 6 sites with elevation between 2-3 km. With Mauna Loa and Mexico_City already being excluded from regional verification evaluation, that left only 9 sites with elevation greater than 2 km being used in the regional evaluations (see new Table S1). Although the RAs tend to have high AOD bias over these mountainous sites, excluding them from the

regional analysis changes little of the verification result (i.e. regional bias, rmse, and r2). These sites are kept in the analysis. In addition, we think the models being unable to represent the topography or sharp elevation gradient due to its coarse resolution is a very possible reason for the models high AOD bias over Mauna Loa. For the 1x1 deg lat/lon model grid where Mauna Loa is located, the topographic height for the one grid is only ~360m (for example for NAAPSRA). This would set the grid elevation within the marine BL in general, while in real world, Mauna Loa AERONET site is 3402 meter above sea level, at about the top of a cone-shaped volcano. So the AERONET site most likely samples free tropospheric AOD, while in the models the grid AOD is affected by both BL and free tropospheric aerosols.

*Line 543 Figure 5: it is different to see the coarse-mode biases. If you single out a different shade (here lighter color for CM) then use to for total and give FM and CM the same shade.*
Reply: I am not sure I totally follow the suggestion. However we tested using other color scales but none of them is better than the current one. So we are keeping the current color for CM, FM and total.

*Figures 6/7: I need more help with the RMS definition. They probably involve data from the same model for all 20 years and also 12 months? It would be more meaningful to have the average seasonality removed, at least for regions with stronger seasonality … Anyway, I would move figure 6 and 7 in the supplement.*
Reply: RMSE is root mean square error. Its definition is expressed as a formula in the appendix B now. Bias, RMSE and $r^2$ are different measures of model performance in the validation effort. We keep all of them in the paper to give a comprehensive view of model performance. The calculation of RMSE is based on 11 year's monthly data for each model. The purpose of the paper is to evaluate the RAs for climate studies, including the seasonality for different regions, so seasonal cycle is something we evaluate and it is not removed. Also, use seasonal cycle based on AERONET data can be biased towards the years with sufficient observational data, while interannual variability of AOD could be large.

*Line 631 It might be nice to show regional component mixtures not annually but for all four seasons*
Reply: We have now included speciated AOD contributions to the total AOD for the four seasons in addition to the annual mean in Figure 10.

*Line 661 This point certainty important. A total bias correction can worsen a (more) skillful component distribution of the forecast model. Even though the maps are illustrative and instructive, these component distributions are not free of error. This is also the reason, why skill I best tested for total AOD.*
Reply: Indeed.

*Line 719 It is nice to offer data access to the four assimilations, but why is a weblocation of MRC data missing?*
*Reply: The web link to the MRC data is now available and included in the manuscript.*

---

## Author Comment (AC2)

Reply to Reviewer Comment 2 on "**Intercomparison of Aerosol Optical Depths from four reanalyses and their multi-reanalysis-consensus**"

Review comments are in italic font, and our replies are in regular font.

*This paper is comparing and analyzing the difference of aerosol reanalyzes (RA) from NAAPS-RA, JRAero, NASA MERRA-2, CAMSRA and a consensus of these for RAs named multi-analyisis consensus (MRC). The parameter used for the comparisons are the AOD (Aeorosol Optical Depth), and the FM (fine mode) part of the AOD, ad the CM (Coarse mode) part of the AOD. The four RAs are compared to each other and the RMC is confronted to all four methods also. As evaluation, AODs from these RAs are compared with AEROsol Robotic NETwork (AERONET) and with the combined MODIS Dark Target/Deep Blue retrievals.*

*The paper is well written, in perfect English, all the parts are very clear.*

*The paper is brilliant in showing the differences between the four RAs and the RMC, also in showing, commenting and analyzing the trends and seasonal and geographical variabilities of AOD, FM-AOD an CM-AOD.*

*The paper suffers of a lack of an explanation concerning the methods. Especially in introduction and part 2. We would expect a longer explanation about what is meant with the "consensus" (RMC). The RMC should be better explained in details: Is it an average of all four other methods? How it differs to a simple average, are there situations/regions/seasons/periods for which one method will have a larger or a weaker weight in the consensus compared to others? If yes why? -> A real effort should be done for explaining better the methods and specificities of this RMC "consensus". Also, you should explain what are the needs of this consensus. Are the four RAs methods not enough?*

*Other lack of explanation in the methods is for the method of comparison AERONET vs. RAs and RMC. The comparison method should here be much better explained (Part 3.3). Especially for the regional comparison: Which stations are taken account for a given region? Is the AERONET value used in the comparison for the region an average of all stations present in the region and accepted for the comparison regarding the criteria explained in Part 2.5? If yes, is it a relevant method = are all n AERONET sites of the region representative with the same weight 1/n for this region? How do you deal then with sub-regions with higher concentrations of AERONET stations? Are there then not over-weighted in the computation of the "regional AERONET-AOD"? Do you have considerations concerning urban and rural AERONET sites?*

Reply: We thank the reviewer's comments. We have followed the constructive comments and revised the manuscript. The "consensus" (RMC) is better and explicitly explained in the introduction, abstract and especially the data/method sections.

To better describe the method of verification of the RAs and MRC with AERONET, we've updated Table S1 to list sites for each region, with latitude/longitude and elevation information of all sites. We've added the following description about how to derive regional validation statistics at the end of section 2.5 "For every AERONET site, the time series of monthly modal AOD from each RA is first extracted from the model grid that encompasses the site's location. Bias, root mean square error (RMSE), and coefficient of determination ($r^2$) are then computed for each site and each RA. The regional validation outcome is derived from the average of validation statistics across all sites within the region (see Table S1 for the sites included in each region). Following the criteria for site selection outlined in section 2.3, only 200 sites are available globally, and certain regions have only a few sites (a minimum of three sites, such as in South Africa) to represent the entire region; hence, no site weighting within a region is applied. It is acknowledged that this averaging method could bias the global validation result toward regions densely populated with sites, notably North America and Europe. The AOD validation results for total, FM, and CM AOD at 550nm are presented accordingly."

To address the urban versus rural question, we've added a new section 3.5 to analyze the difference in the performance of these RAs in terms of modal AODs compared to AERONET.

*These for the general scientific comments.*
*A general presentation comment is that they are a lot of acronyms used. Positive is that the authors very often detailed in the text several times the meaning of the acronym. Nevertheless, I suggest you to add an acronym table in order to help the reader to understand quickly the acronyms.*

Reply: Thanks for the suggestion. We've added an acronym table in Appendix A, and definition of terminologies in Appendix B.

*I have some specific comments/questions for two pictures:*

*- Maps of Figure 1: First line of maps (MODIS): I understand that on the Antarctic region during JJA there are no measurements, but why is it the same during the other seasons (example austral summer in DJF)?*

Reply: We've added "In the MODIS plots, the white area means a lack of data attributed to either none valid-retrievals or quality-control filtering" in the figure caption for Figure 1. The MODIS AOD product we used is a data-assimilation quality product, which has a cut-off at 40°S to filter out potential cloud-contaminated data south of this latitude. This is added in the data description in Section 2.4.

*- Figure 10: 10b) On the Antarctic Graphic: It is so obvious that JRAero is much more overestimating the AOD that it is worth that you give an explanation in a comment in the text of the article*

Reply: We've added in the Figure 10 result "The total AOD in JRAero exhibits exceptionally high levels, primarily attributed to elevated sea salt and sulfate AODs (Fig. S5). This anomaly stems from the MASINGAR model used to produce JRAero, which tended to underestimate the removal of aerosols via cumulus convection. Consequently, this led to an overestimation of aerosol concentrations in the polar regions and the upper atmosphere. The underestimation of the removal process has been resolved in the current MASINGAR model and the overestimation of AOD over the polar regions will be improved with the JRAero version upgrade. "

*And a comment about References:*
*- The most important cited reference (Sessions et al. 2015), cited L104, L235, L292, L351 is missing in the "References" part.*

*- Sessions et al. 2016 (cited in L66-67) is missing in the References list*

Reply: Thank you for bringing this to our attention. We have included the Sessions et al., 2015 reference in the reference list. Regarding "Sessions et al. 2016", the year "2016" was indeed a typo. It has been corrected to "2015" now.